# FULLPART: GENERATING EACH 3D PART AT FULL RESOLUTION

**Lihe Ding**[1*], **Shaocong Dong**[2*], **Yaokun Li**[1], **Chenjian Gao**[1], **Xiao Chen**[1], **Rui Han**[3], **Yihao Kuang**[4]
**Hong Zhang**[4], **Bo Huang**[4], **Zhanpeng Huang**[3], **Zibin Wang**[3], **Dan Xu**[2†], **Tianfan Xue**[1,5†]
[1]CUHK MMLab; [2]HKUST; [3]SenseTime Research; [4]Chongqing University; [5]CPII under InnoHK
{dl023, tfxue}@ie.cuhk.edu.hk; {sdongae, danxu}@cse.ust.hk

## ABSTRACT

Part-based 3D generation holds great potential for various applications. Previous part generators that represent parts using implicit vector-set tokens often suffer from insufficient geometric details. Another line of work adopts an explicit voxel representation but shares a global voxel grid among all parts; this often causes small parts to occupy too few voxels, leading to degraded quality. In this paper, we propose *FullPart*, a novel framework that combines both implicit and explicit paradigms. It first derives the bounding box layout through an implicit box vector-set diffusion process, a task that implicit diffusion handles effectively since box tokens contain little geometric detail. Then, it generates detailed parts, each within its own fixed full-resolution voxel grid. Instead of sharing a global low-resolution space, each part in our method—even small ones—is generated at full resolution, enabling the synthesis of intricate details. We further introduce a center-point encoding strategy to address the misalignment issue when exchanging information between parts of different actual sizes, thereby maintaining global coherence. Moreover, to tackle the scarcity of reliable part data, we present *PartVerse-XL*, the largest human-annotated 3D part dataset to date with 40K objects and 320K parts. Extensive experiments demonstrate that FullPart achieves state-of-the-art results in 3D part generation. Code, model, and dataset are available at https://fullpart3d.github.io.

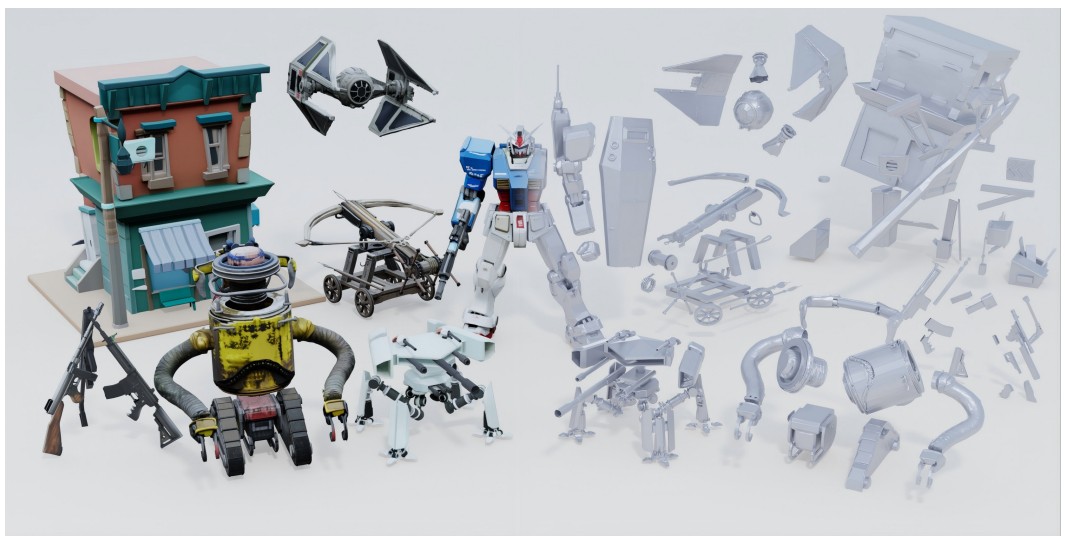

Figure 1: FullPart achieves high-quality part-based 3D generation.

---

*Equal contribution †Corresponding authors

# 1 INTRODUCTION

Part-based 3D generation and manipulation have wide applications in virtual reality, gaming, robotics, and digital content creation. While recent neural 3D generation methods have demonstrated impressive results in 3D object synthesis (Zhang et al., 2024; Xiang et al., 2025; Li et al., 2025; Zhao et al., 2025), the majority of methods did not provide detailed part decomposition, which is essential for downstream tasks such as texture mapping, animation, physical simulation, or fine-grained editing processes. Ideally, a more useful 3D generation framework shall also capture the compositional nature of real-world objects. It shall decompose the generated object into semantically meaningful parts, and users can manipulate each part independently while global coherence can be maintained automatically.

To achieve this, researchers recently proposed part-level synthesis methods, primarily focusing on two paradigms. One paradigm employs implicit latent part-based representations (Lin et al., 2025; Chen et al., 2025b; Tang et al., 2025), where each part corresponds to an independent set of latent tokens jointly generated by a shared model. This approach benefits from end-to-end training that simultaneously learns both part geometry and spatial layout, simplifying the training pipeline. However, this implicit representation i) suffers from insufficient part details due to the limited query resolution when decoding part vecsets and ii) cannot precisely model spatial mappings, making it suboptimal for texture generation, multi-modal applications, or precise 3D editing. Another paradigm by Yang et al. (2025b) explicitly defines part layer using bounding boxes, and generates detailed voxel structures within each box. While this approach is good for layout modeling, it is challenging to fine detail generation and maintain the global structural coherence, particularly when handling complex or intricately connected components. Besides, all existing approaches share a critical limitation: both solutions force all parts to share a single global representation space, which limits the resolution allocated to each part and results in poor details of small but complex 3D parts.

In this work, we present two key insights: i) while implicit representations struggle with fine part details, they are well-suited for generating layouts that contain only bounding box information without geometric detail; and ii) explicit representations should allocate an isolated full-resolution space to each part; otherwise, small parts may occupy only a few voxels, resulting in degraded quality.

Based on the these observations, we present FullPart, a novel 3D part generation framework that first derives layouts (bounding boxes) from implicit vecset generation and then generates each part at full resolution with explicit representation. In this way, we combine both implicit and explicit paradigms while addressing their respective limitations. Specifically, FullPart follows a three-stage generation process: (1) layout generation by representing bounding boxes with latent vecsets; (2) dividing each box into an isolated $N^3$ grid and generating coarse 3D part structure with full resolution; and (3) refinement of textured meshes based on the coarse structural foundation.

One remaining challenge of this design is to maintain part coherence when assigning each part to an isolated full-resolution grid. Since each part is generated in its own fixed grid space ($64^3$ in our setup), there is a resolution mismatch problem for tokens on the boundary of two neighboring parts with different sizes: tokens (voxels) from the larger part contain fewer details and tokens from the smaller part contain more finer details. This may result in artifacts in overlapping regions when stitching two parts together. Moreover, since tokens from different parts correspond to voxels of varying spatial sizes within the global coordinate system, directly applying attention mechanisms to exchange information between part tokens would result in scale misalignment. To address these issues, we propose a specialized center-corner encoding mechanism. For each part, we calculate the positional embeddings of all eight corners and inject them as well as the centers into all tokens in this part. In this way, each token is aware of its actual spatial extent in the voxel grid, and based on which, diffusion will learn to stitch different parts smoothly. Additional analysis also shows this also makes finetuning easier, which we will further discuss in Section 3.3. With this design, the generated 3D parts can be smoothly stitched together, as shown in Figure 1.

Furthermore, to support high-quality part generation, we introduce PartVerse-XL, the largest and most comprehensively annotated 3D part dataset, consisting 40K objects and 320K parts, with associated part-aware texture descriptions. The reason we built this new dataset is that existing 3D datasets either have no 3D part labeling or very few objects have part labels, or the label quality is not high enough. Some 3D model metadata contains part information from artists' modeling processes, these annotations are often incomplete and lack semantic consistency (e.g., some artists may

treat object skin as a separate part). Therefore, we selected 40K objects from Objaverse-XL (Deitke et al., 2023), and create a high-quality part annotation, using mesh pre-segmentation followed by human refinement. We will release this dataset to benefit future research in part generation.

Through extensive experimentation, we demonstrate that FullPart achieves superior performance in both part-level fidelity and global structural coherence compared to existing approaches, with particular strength in generating plausible geometries for occluded and small parts.

In summary, our contributions are: i) We propose FullPart, a novel part-level 3D generation framework that combines implicit layout representation with explicit part structure generation, enabling precise detail control; ii) We enable each part to be generated in an isolated full resolution while maintaining the global part coherence and preventing violation of the foundation model's established knowledge by proposing a center-corner encoding strategy; iii) We present PartVerse-XL, a large-scale human-annotated part dataset containing 320K high-quality parts with part-aware textual descriptions, addressing the scarcity of reliable part-level 3D training data; iv) We demonstrate state-of-the-art performance in part-level 3D generation across multiple metrics, with particular strength in handling complex part interactions and generating plausible occluded geometries. Code, model, and dataset are available at `https://fullpart3d.github.io`.

## 2 RELATED WORK

### 2.1 3D GENERATION

Early 3D generation focused on category-specific or image-to-3D approaches with limited object diversity (Poole et al., 2022; Wang et al., 2023; Lin et al., 2023; Dong et al., 2024; Hong et al., 2023; Liu et al., 2023a; Shi et al., 2023; Ding et al., 2024). This paradigm shifted with 3D-native diffusion models operating directly in 3D space. CLAY (Zhang et al., 2024) established a foundational transformer-based architecture for direct 3D generation. Recent advances (2024–2025) significantly improved fidelity and control: TRELLIS (Xiang et al., 2025) introduced sparse voxel-based structured latent representations for precise geometry; TripoSG (Li et al., 2025) leveraged SDF representations with rectified flow for speed-quality trade-offs; Direct3D (Wu et al., 2025) enabled high-resolution text-to-3D generation; and Hunyuan3D (Zhao et al., 2025) integrated multi-modal understanding for text-guided manipulation. However, these frameworks generate monolithic shapes without explicit part decomposition, limiting fine-grained editing.

### 2.2 PART GENERATION

The need for part-aware 3D generation has motivated several research directions with different 3D representations and learning paradigms. Early part-aware approaches typically relied on category-specific annotations and auto-encoder-style architectures, such as SPAGHETTI Hertz et al. (2022) (implicit representation) and Neural Template Hui et al. (2022) (implicit and explicit), or focused on diffusion-based part generation, including SALAD Koo et al. (2023) (implicit) and DiffFacto (Nakayama et al., 2023) (implicit). However, these methods suffer from limited generalizability across diverse object categories. Subsequent works such as Part123 Liu et al. (2024) and PartGen Chen et al. (2025a) used SAM Kirillov et al. (2023) for multi-view segmentation, yet remained constrained by segmentation quality and limited patch information. Recent contemporaneous works have made significant strides toward general-purpose part generation. CoPart (Dong et al., 2025) represented a 3D object with multiple contextual part latents and simultaneously generated coherent 3D parts. HoloPart (Yang et al., 2025a) segmented objects into parts and then completed the segments into full 3D parts. PartCrafter (Lin et al., 2025) introduced an implicit latent representation approach where each part corresponds to an independent set of latent tokens generated by a shared model. AutoPartGen (Chen et al., 2025b) automated part decomposition through a learnable part proposal mechanism. OmniPart (Yang et al., 2025b) adopted explicit voxel-based representations with bounding boxes to define part layouts. BANG (Zhang et al., 2025) represented 3D objects as a smooth sequence of exploded states and generated parts by generative exploded dynamics.

However, all existing part-aware generation methods share a critical limitation: they force parts to share a single global representation space. In implicit representation approaches, this means small parts receive insufficient representation capacity in the shared latent space. In voxel-based methods, this results in small parts occupying only a tiny fraction of a shared N×N×N grid, leading to extremely low effective resolution for those components. This fundamental bottleneck has not been

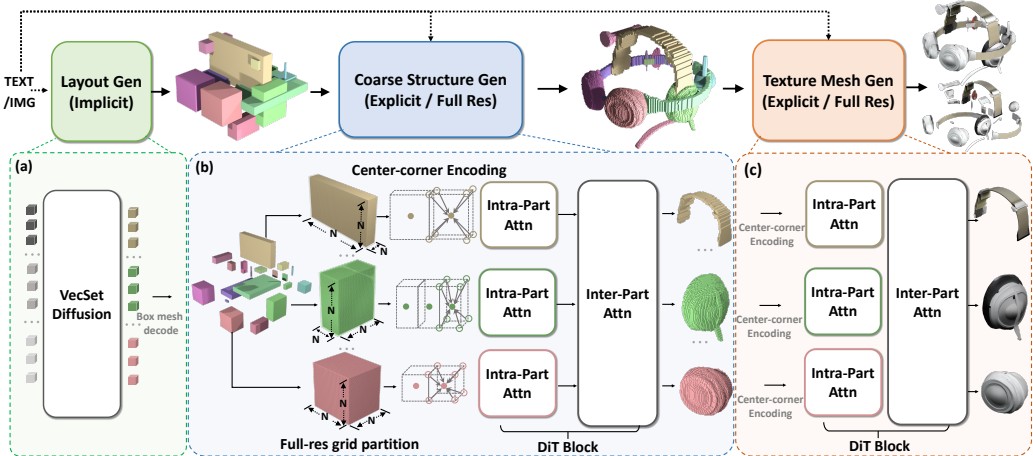

Figure 2: FullPart framework. FullPart comprises three sequential stages: (a) layout generation using implicit vecset diffusion, (b) generating each part at a full-resolution grid with explicit voxel representation, and (c) refining coarse part structures to texture meshes.

adequately addressed by prior work, which is precisely where our framework makes its key contribution by treating each part as a full-resolution independent object during the generation process.

## 2.3 DECOMPOSABLE 3D GENERATION

Decomposable generation is crucial for 3D scene and avatar synthesis. Frankenstein Yan et al. (2024) encoded input into a compact triplane and generated semantic-compositional 3D scenes or avatars via a triplane diffusion process conditioned on a 2D semantic layout. LayoutGPT Feng et al. (2023) predicted layout parameters (e.g., bounding boxes, categories) from an LLM and directly retrieved objects. CAST Yao et al. (2025) achieved component-aligned 3D scene generation from a single RGB image by VLM scene analysis, pose-aware generation module, and physical constraint refinement. StdGEN He et al. (2025) generated 3D avatar implicit fields through a semantic-aware large reconstruction model and further employed a multi-layer semantic surface extraction scheme together with a refinement module to obtain the decomposable meshes.

## 3 METHODOLOGY

We formally define our part-aware 3D generation problem as follows: given a conditioning input (typically a single-view RGB image or a text prompt), our goal is to generate a structured 3D object $\mathbf{O} = \{\mathbf{o}_i\}_{i=1}^K$ consisting of $K$ semantically meaningful parts, where each part $\mathbf{o}_i$ is represented as a textured mesh with explicit geometric and topological properties. Unlike monolithic 3D generation approaches, our framework explicitly models the compositional nature of objects through a hierarchical generation process that first establishes part layouts (bounding boxes) using implicit vecset diffusion (Li et al., 2025) (Figure 2 (a)), then generates coarse part structures within boxes by representing each part in an explicit full-resolution voxel grid (Figure 2 (b)), and finally refines coarse voxels into detailed meshes with textures (Figure 2 (c)). In the remainder of this section, we provide a comprehensive description of each component.

### 3.1 PRELIMINARY: 3D OBJECT GENERATION FRAMEWORKS

Our methodology builds upon two predominant paradigms in 3D generation: implicit latent representations for layout generation and explicit voxel-based representations for part structure generation. We briefly review these approaches and establish the foundation for our part-aware extension.

**Implicit Representations.** Following 3DShape2VecSet (Zhang et al., 2023), a 3D object can be represented as a vecset—a set of latent tokens $\mathbf{T} = \{\mathbf{t}_j\}_{j=1}^M \in \mathbb{R}^{M \times D}$, where $M$ is the number of tokens and $\mathbf{t}_j \in \mathbb{R}^D$ is the $j$-th token with feature dimension $D$. The decoder transforms these tokens into a signed distance field (SDF) representation $\phi : \mathbb{R}^3 \to \mathbb{R}$, where the zero level set defines the object surface: $\mathcal{S} = \{\mathbf{x} \in \mathbb{R}^3 | \phi(\mathbf{x}) = 0\}$. This implicit representation enables high-fidelity geometry generation but lacks explicit spatial partitioning for part manipulation.

**Explicit Representations.** TRELLIS (Xiang et al., 2025) introduces a structured latent representation for 3D objects through sparse voxels. Given a 3D asset, it encodes geometric information into a set of active featured voxels $\mathbf{C}$:

$$\mathbf{C} = \{\mathbf{c}_i | \mathbf{c}_i = (\mathbf{f}_i, \mathbf{p}_i)\}_{i=1}^{L}, \quad \mathbf{f}_i \in \mathbb{R}^D, \quad \mathbf{p}_i \in \{0, 1, \ldots, N-1\}^3, \tag{1}$$

where $\mathbf{p}_i$ denotes the positional index of an active voxel in the $N \times N \times N$ grid, and $\mathbf{f}_i$ represents the feature vector capturing local geometry and appearance. The active voxels $\mathbf{p}_i$ define the coarse structure, while $\mathbf{f}_i$ encodes fine details.

**Attention Mechanisms for Part Generation.** Building upon the previous works like CoPart (Dong et al., 2025), we introduce two attention mechanisms critical for part-aware generation:

*Intra-Part Attention:* For part $k$, given its token set $\mathbf{T}_k \in \mathbb{R}^{M \times D}$, intra-part attention limits the self-attention computation to the tokens of part $k$:

$$\mathbf{Q}_k = \mathbf{W}_q \mathbf{T}_k; \ \mathbf{K}_k = \mathbf{W}_k \mathbf{T}_k; \ \mathbf{V}_k = \mathbf{W}_v \mathbf{T}_k \tag{2}$$

where $\mathbf{Q}_k, \mathbf{K}_k, \mathbf{V}_k$ are query, key, and value projections of $\mathbf{T}_k$.

*Inter-Part Attention:* Given all part tokens $\mathbf{T} = [\mathbf{T}_1, \ldots, \mathbf{T}_K] \in \mathbb{R}^{KM \times D}$, inter-part attention computes self-attention between all tokens:

$$\mathbf{Q} = \mathbf{W}_q \mathbf{T}; \ \mathbf{K} = \mathbf{W}_k \mathbf{T}; \ \mathbf{V} = \mathbf{W}_v \mathbf{T} \tag{3}$$

where $\mathbf{Q}, \mathbf{K}, \mathbf{V}$ are projections of the concatenated token set $\mathbf{T}$.

These attention mechanisms enable our framework to balance intra-part detail generation with inter-part structural coherence. The flexibility of transformer architecture allows seamless adaptation of full-object generation models to part-level synthesis while preserving pre-trained model priors.

### 3.2 LAYOUT GENERATION

Our layout generation module produces a set of bounding boxes that define the spatial arrangement of object parts. Rather than treating boxes as abstract parameters, we represent each box $\mathbf{b}_k$ as a minimal triangular mesh (a cuboid with 8 vertices and 12 faces), where the collection of these meshes forms a coarse "blocky" representation of the object, reminiscent of Minecraft-style models. By representing bounding boxes as meshes, we obtain a semantic representation aligned with the latent space of the vecset diffusion model, thereby leveraging its strong prior for effective layout generation.

Formally, given a conditioning image or text prompt, we generate a set of $K'$ bounding box meshes $\mathbf{B} = \{\mathbf{b}_k\}_{k=1}^{K'}$, where $\mathbf{b}_k = (\mathbf{v}_k, \mathbf{f}_k)$ is the $k$-th box mesh with cuboid vertices $\mathbf{v}_k \in \mathbb{R}^{8 \times 3}$ and face indices $\mathbf{f}_k \in \mathbb{N}^{12 \times 3}$.

To encode these box meshes, we utilize the VAE from TripoSG (Li et al., 2025), which maps each box to $M$ latent tokens: $\mathbf{T}_k = \text{VAE}_{\text{enc}}(\mathbf{b}_k) \in \mathbb{R}^{M \times D}, \quad k = 1, \ldots, K'$.

To make the network distinguish tokens from different boxes, we inject box ID embeddings $\mathbf{e}_{\text{id}}(k) \in \mathbb{R}^D$ to the corresponding token $\mathbf{t}_k \in \mathbb{R}^D$ (tokens from box $k$): $\tilde{\mathbf{t}}_k = \mathbf{t}_k + \mathbf{e}_{\text{id}}(k)$.

Additionally, to leverage the powerful priors of the foundation model, we retain a global branch in the original vecset diffusion model, which is tasked with predicting the holistic object structure. This global branch provides essential semantic guidance for the layout generation process. To enable the network to distinguish its tokens from part-specific box tokens, we assign an identifier of 0 to the tokens $\mathbf{T}_0 \in \mathbb{R}^{M \times D}$ in the global branch.

During training, for objects with $K < K'$ actual boxes, we pad the token sequences with zero vectors for the remaining $K' - K$ boxes. The complete token sequence that represents the box layout for the DiT input is: $\mathbf{T}_{\text{all}} = [\mathbf{T}_0, \mathbf{T}_1, \ldots, \mathbf{T}_{K'}] \in \mathbb{R}^{(K'+1)M \times D}$, where $T_0$ is the tokens from the global branch. Our DiT employs a hybrid attention strategy: intra-part attention operates within each box's token set ($\mathbf{T}_k$ for $k = 0, 1, \ldots, K'$), while inter-part attention operates across all box tokens. This enables the model to learn both local-specific characteristics and global structural relationships.

At inference time, we sample the latent tokens $\mathbf{T}_{\text{all}}$ using a diffusion process, then decode them back to box meshes via the VAE decoder: $\hat{\mathbf{B}} = \{\text{VAE}_{\text{dec}}(\mathbf{T}_k)\}_{k=1}^{K}$. Finally, due to potential deformation

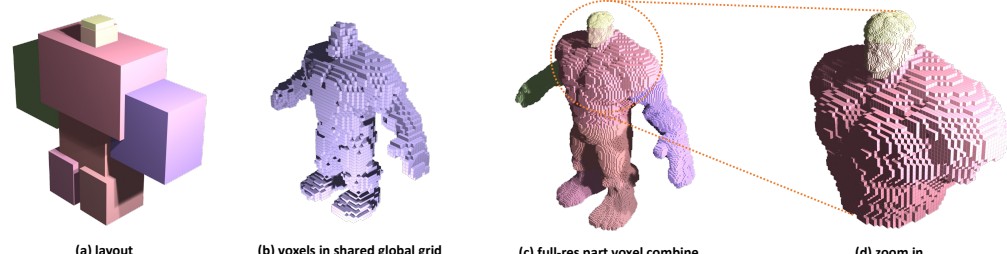

|  |  |  |  |
|---|---|---|---|
| **(a) layout** | **(b) voxels in shared global grid** | **(c) full-res part voxel combine** | **(d) zoom in** |

Figure 3: Illustration of our 3D part representation. Our model generates each part at isolated full resolution (c), which contains more fine details than the previous sharing global voxel grid strategy (b). Also, tokens from different parts represent varying spatial extents, e.g., head and body in (d).

and incorrect shapes with the decoded meshes $\hat{\mathbf{B}}$, we recalculate their own bounding boxes $\hat{\mathbf{B}}'$, and retain those with high IoU between corresponding $\hat{\mathbf{B}}$ and $\hat{\mathbf{B}}'$ as layout inputs for subsequent stages.

### 3.3 3D COARSE STRUCTURE GENERATION

Given the bounding boxes from the layout generation, our coarse structure generation stage creates detailed part geometries within the layout-defined regions. Instead of sharing a global voxel grid (Figure 3 (b)), we generate each part within its own dedicated $N \times N \times N$ voxel space at full resolution, enabling precise detail generation regardless of part size, as shown in Figure 3 (c).

To generate each part at a full resolution, we normalize it to the canonical space $[-1, 1]^3$ and define a binary occupancy grid: $\mathbf{V}_k \in \{0, 1\}^{N^3}$, where $\mathbf{V}_k[x, y, z] = 1$ if occupied, and $k$ denotes the $k$-th part. This normalization ensures that even small parts utilize the full resolution of the voxel grid, overcoming the resolution limitations of shared-grid approaches. However, when each part is defined in its own full-resolution grid, the tokens from different bounding box regions represent varying spatial extents. This discrepancy becomes problematic when exchanging information between parts. For example, a token from a big part (e.g., upper body in Figure 3 (d)) may represent an absolute voxel size several times larger than a token from a small part (e.g., head in Figure 3 (d)), leading to token misalignment.

To address this, we introduce a *center-corner encoding mechanism* that embeds the absolute spatial context of each voxel. For a voxel at position $\mathbf{u} = (x, y, z)$ in the normalized grid of part $k$, we compute its 8 corresponding corners in the global object space: $\{\mathbf{u}_g^i | \mathbf{u}_g^i = \mathcal{T}(\mathbf{u}^i, \mathbf{b}_k)\}_{i=0}^{7}$, where $\mathcal{T}(, \mathbf{b}_k)$ is the transformation to global coordinate using box $\mathbf{b}_k$ and $\mathbf{u}_g^i$ is the $i$-th corner of $\mathbf{u}$ in global coordinate. Following the foundation model that encodes integer positions in the global coordinate, we partition global space into a super-high resolution grid ($2048 \times 2048 \times 2048$) and find the integer coordinates of all eight corners $\{\lfloor \mathbf{u}_g^i \rfloor\}_{i=0}^{7}$ and the center $\lfloor \mathbf{u}_g \rfloor$. At this stage, we have obtained the center and eight corner coordinates for each voxel of every part. Although different voxels may represent different spatial extents, their corner and center positions are now expressed within a unified super-high-resolution global coordinate system. This allows us to encode these positions using the pre-trained positional embedding layer directly.

One remaining consideration is that our model is based on a pre-trained 3D generator, whose positional encoding layers were only trained under a low resolution of $64 \times 64 \times 64$. Fortunately, previous research has shown that positional encoding can be effectively extrapolated during fine-tuning (Liu et al., 2023b). Therefore, we simply inject the positional embeddings of one center and eight corners for each token: $\mathbf{t}_{\mathbf{u}}^k = \mathbf{e}_{\text{pos}}(\lfloor \mathbf{u}_g \rfloor) + \sum_{i=0}^{7} \mathbf{e}_{\text{pos}}(\lfloor \mathbf{u}_g^i \rfloor) + \mathbf{e}_{\text{id}}(k)$, where $\mathbf{e}_{\text{pos}}()$ is the positional embedding layer and $\mathbf{e}_{\text{id}}()$ is an additional embedding layer to inject part ID information. This center-corner embedding is added to each voxel token, providing explicit spatial context that enables the model to understand part relationships despite the normalization. Furthermore, our strategy requires no modification to the pre-trained model's architecture, thus it can better utilize its foundational priors. Similar to the layout stage, we include a global branch ($k = 0$) that processes the entire object structure. The complete token sequence is: $\mathbf{T}_{\text{all}} = [\mathbf{T}_0, \mathbf{T}_1, \ldots, \mathbf{T}_{K'}]$.

Our DiT architecture employs the same hybrid attention mechanism as in the layout stage, allowing it to capture both fine-grained part details and global structural coherence. The diffusion process generates the voxel tokens conditioned on the input image or text prompt, which is incorporated through cross-attention layers following TRELLIS's conditioning strategy.

## 3.4 REFINEMENT

The refinement stage enhances the coarse voxel structures with detailed geometry and textures, adapting TRELLIS's second stage (Xiang et al., 2025) for part-aware generation. For each part $k$, we obtain feature vectors $\mathbf{F}_k \in \mathbb{R}^{L_k \times D_f}$ for the occupied voxels $\mathcal{P}_k$ via TRELLIS's VAE encoder, where $D_f$ is the feature dimension.

The VAE encoder embeds the tokens with both structure and projected multi-view features (from Dino-v2 by Oquab et al. (2023)). A key challenge is handling occlusions between parts. To address this, we normalize and render each part independently, obtaining tokens with part-specific image features, as the denoising target during training. However, during inference, our model only requires a single global image or text as a conditional input, making it practical for real-world applications.

The diffusion model generates the feature vectors $\mathbf{F}_k$ on all occupied voxels $\mathcal{P}_k$ at full resolution. These features are then decoded into textured meshes using TRELLIS's decoder: $\hat{\mathbf{o}}_k = \text{Decoder}(\mathbf{F}_k, \mathcal{P}_k)$. The final object is assembled from the individual part meshes: $\hat{\mathbf{O}} = \bigcup_{k=1}^{K} \hat{\mathbf{o}}_k$.

## 3.5 OPTIMIZATION LOSS

All stages of our framework are trained using Conditional Flow Matching (CFM) (Lipman et al., 2023) objective, i.e., for a given stage with token representations $\mathbf{x}$, we have the training objective:

$$\mathcal{L}_{\text{cfm}}(\boldsymbol{\theta}) = \mathbb{E}_{\mathbf{x}_0, \epsilon, t} \left[ \| \mathbf{v}_\theta(\mathbf{x}, t) - (\epsilon - \mathbf{x}_0) \|_2^2 \right]. \tag{4}$$

where $\epsilon_\theta$ is the diffusion noise, and $\mathbf{v}_\theta$ is the predicting vector field.

# 4 PARTVERSE-XL DATASET

To support large-scale part-aware 3D generation, we introduce *PartVerse-XL*, an expanded and refined extension of *PartVerse* (Dong et al., 2025). It contains 40K high-quality 3D objects from *Objaverse-XL* (Deitke et al., 2023), yielding 320K semantically consistent, textured parts across over 200 categories, each paired with a descriptive caption. This scale and diversity significantly surpass prior benchmarks and enable robust training of models requiring fine-grained part semantics and geometry.

We construct *PartVerse-XL* via a two-stage pipeline. First, we apply an automated pre-segmentation algorithm that fuses geometric priors (e.g., mesh connectivity, UV seams) with semantic cues from SAM-2 (Kirillov et al., 2023) and Samesh (Tang et al., 2024), deliberately producing over-segmented outputs for easier human correction. Second, expert annotators refine the segments using a Blender-based tool—merging or splitting components to ensure semantic clarity, structural symmetry, and texture preservation—while discarding low-quality or ambiguous assets.

For each part, we generate a textual caption by rendering multi-view images of both the full object and the isolated part. We select the view with maximal part–whole visibility overlap, overlay a bounding box around the part, and feed the composite image to a vision-language model. Captions are generated to accurately describe shape, appearance, material, and part–object relationships (e.g., "a cylindrical metallic handle attached to the right side of a coffee mug").

*PartVerse-XL* establishes a new standard in scale, semantic fidelity, and multimodal alignment for part-level 3D generation. More details of our dataset can be found in the supplementary material.

# 5 EXPERIMENTS

This section presents our experimental results. All experiments of our model utilize the PartVerse-XL training set (40K objects, 320K parts), and we also construct a dedicated test set of 100 objects. More details, results, and analysis can be found in the supplementary materials.

## 5.1 IMPLEMENTATION DETAILS

We train FullPart in three sequential stages on 8 NVIDIA A100 GPUs. The layout generator (Stage 1) is trained for 96 hours using AdamW ($\beta_1 = 0.9$, $\beta_2 = 0.999$) with batch size 64. Stages 2 (coarse voxel generation) and 3 (mesh refinement) each train for 144 hours with batch size 8, leveraging pre-trained TRELLIS Xiang et al. (2025) weights. The maximum part count is clamped to $K_{\max} = 30$, and all parts are generated in isolated full resolution $64^3$ grids. During inference, we apply non-maximum suppression (NMS) with IoU threshold 0.7 to eliminate redundant boxes. Besides, we take 100 manually selected untrained objects as the test set. More details can be found in the supplementary material.

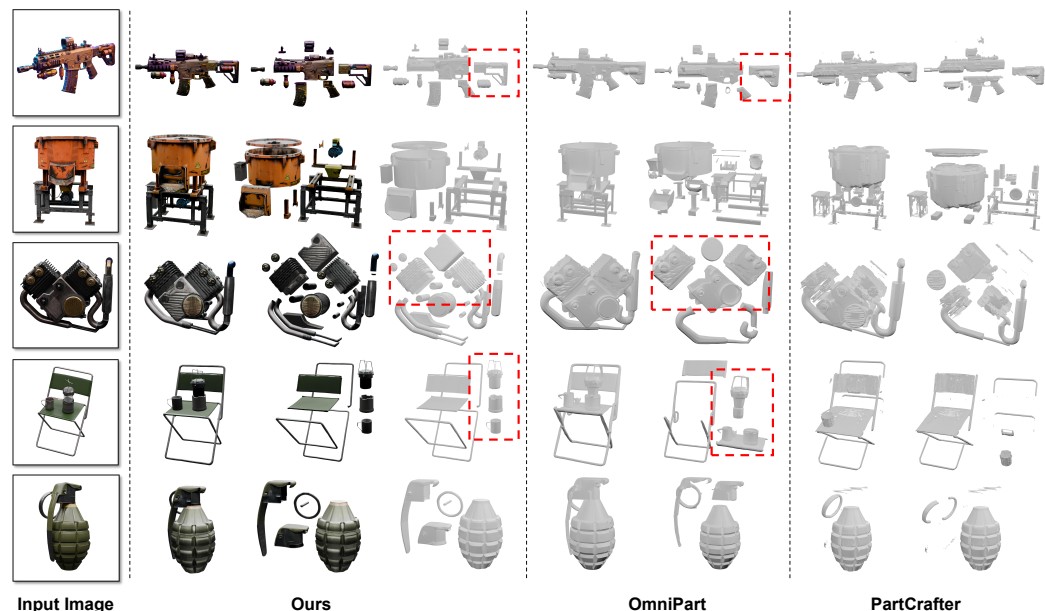

Figure 4: Comparison with state-of-the-art 3D Part generators. Our method can generate more detailed and reasonably divided parts.

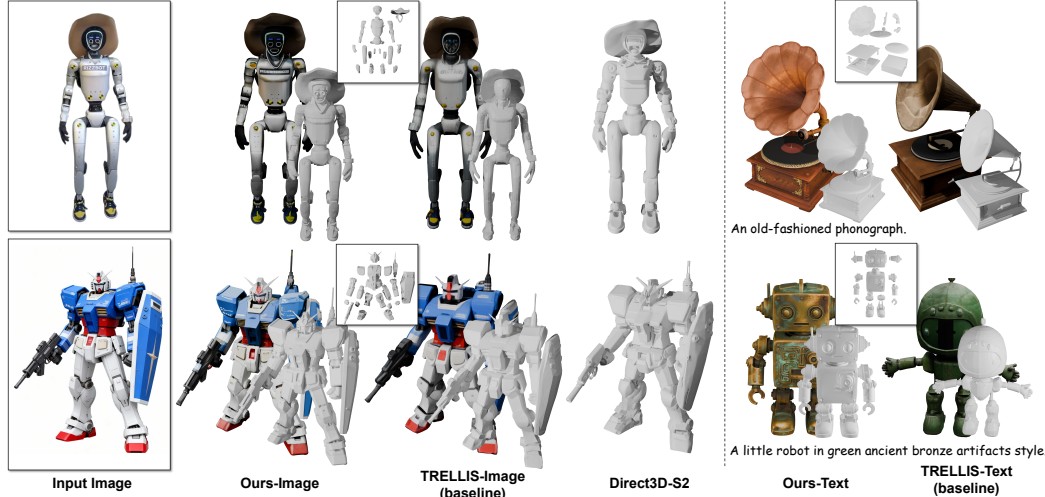

Figure 5: Comparison with the state-of-the-art 3D generators.

## 5.2 RESULTS

### 5.2.1 COMPARISON WITH STATE-OF-THE-ART PART GENERATORS

We compare FullPart against two leading part-aware generators: PartCrafter Lin et al. (2025), and Omnipart Yang et al. (2025b). Figure 4 demonstrates that FullPart achieves superior geometric fidelity and structural coherence, particularly for intricate assemblies (e.g., articulated robot arms) and occluded regions (e.g., chair undersides). While PartCrafter produces fragmented parts due to implicit token entanglement, and Omnipart suffers from voxelization artifacts in small components (e.g., thin chair legs), FullPart preserves fine details through dedicated per-part full-resolution grids.

### 5.2.2 COMPARISON WITH FULL-OBJECT METHODS

We compare FullPart against its foundational model TRELLIS Xiang et al. (2025) and the state-of-the-art monolithic 3D generator Direct3D-S2 Wu et al. (2025). As these full-object approaches lack part decomposition capability, they inherently suffer from global grid sparsity, leading to significant detail loss in fine-grained regions (e.g., robotic head features in Figure 5). In contrast, FullPart's part-aware architecture preserves high-fidelity details through localized high-resolution generation.

Table 1: Quantitative comparison on PartVerse-XL test set ("-" denotes not applicable).

| Method | F-Score ↑ | CD ↓ | Part-CD ↓ | ULIP-Score ↑ |
|---|---|---|---|---|
| TRELLIS Xiang et al. (2025) | 0.71 | 0.16 | - | 0.21 |
| HoloPart Lin et al. (2025) | 0.68 | 0.21 | - | 0.15 |
| PartCrafter Chen et al. (2025b) | 0.63 | 0.42 | - | 0.13 |
| Omnipart Yang et al. (2025b) | 0.77 | 0.15 | 0.42 | 0.22 |
| **FullPart (Ours)** | **0.81** | **0.11** | **0.36** | **0.24** |

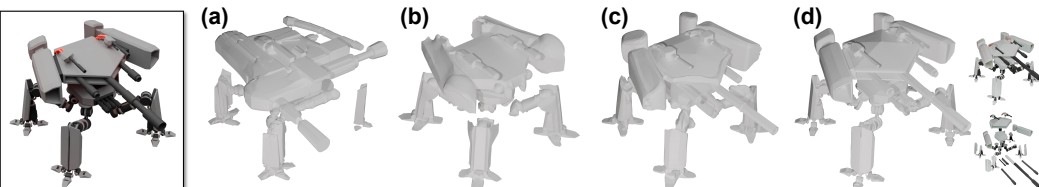

**Input**     **w/o Corner Enc**     **w/o Human Anno**     **w/o Per-part Grid**     **FullPart Baseline**

Figure 6: Comparison with different model settings under identical training budgets: (a) no corner encoding, (b) using metadata-derived structural information without manual annotations, (c) all layout boxes constrained to a single voxel space, causing each box may only occupy a small number of voxels, and (d) normal setting with all things.

### 5.2.3 QUANTITATIVE EVALUATION

Table 1 reports metrics on the 100-object test set. We evaluate: (i) Global fidelity with a threshold of 0.1 (F-Score), (ii) Global mesh chamfer distance (CD), (iii) Part mesh chamfer distance when use same layout boxes (Part-CD), and (iv) 3D Semantic alignment (ULIP Score Xue et al. (2023)). FullPart outperforms all baselines in part-level and full-level metrics, proving its strength in part-level detail and global coherence. Note that Part-CD is not applicable to the first three methods due to their lack of bounding box-conditioned part generation capability.

### 5.3 ABLATION STUDIES

**Center-corner Encoding Ablation.** To validate our center-corner encoding strategy, we ablate the generator by replacing center + corner coordinates with only center coordinates. Figure 6 (a) shows that this variant fails to model part interactions (e.g., misaligned chair legs). This confirms that explicit location and scale information are critical for capturing spatial relationships between parts.

**Impact of Human-post Annotations.** We train FullPart on two data variants: (i) raw metadata (artist-provided part labels), and (ii) PartVerse-XL with human-refined annotations. As Figure 6 (b) (d) illustrates, metadata-only training produces semantically incorrect parts due to the noise in part labels, while training on human-annotated data yields coherent, functionally meaningful parts.

**Per-Part Full-Resolution Grid Ablation.** We compare our dedicated $N^3$ ($N = 64$ in our setting) per-part grids against a global grid baseline where all parts share the global $N^3$ space. Figure 6 (c) reveals severe detail degradation in small parts under the global grid, as they occupy only a small number of voxels. Our method maintains consistent resolution across all parts, achieving uniform detail generation regardless of relative size.

### 5.4 APPLICATION

The FullPart framework exhibits strong flexibility, enabling a range of controllable 3D editing applications through manipulation of the layout boxes. Specifically, users can intuitively edit part-level geometry by adding, deleting, or modifying the shape and position of bounding boxes—each operation directly influences the corresponding generated part while preserving the integrity of the rest of the object. As illustrated in Figure 7, we demonstrate an example where two accessories are added to a rifle and

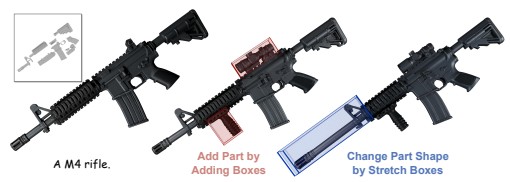

Figure 7: Editing applications.

the barrel is elongated. After the initial generation, we modify the layout boxes by inserting new boxes and stretching existing ones, then re-run inference. For unchanged parts, we bypass the diffusion sampling process by directly injecting their clean latent tokens (along with the appropriate noise level corresponding to the current inference timestep) into the DiT as fixed inputs. This

strategy ensures consistent regeneration of unedited components while allowing efficient, localized updates—highlighting FullPart's suitability for interactive, part-aware 3D content creation.

## 6 CONCLUSION

We introduced FullPart, a part-aware 3D generation framework that integrates implicit and explicit representations. By treating each part as a full-resolution object, our method overcomes the resolution bottleneck of shared voxel grids. We also introduce PartVerse-XL, a large-scale annotated part dataset to advance future research in 3D part generation.

## ACKNOWLEDGEMENTS

The work is supported in part by the Early Career Scheme ofthe Research Grants Council (RGC) of the Hong Kong SARunder grant No. 26202321, ITF PRP/046/24FX, SAIL ResearchProject, HKUST-Zeekr Collaborative Research Fund, and CUHK-CUHKSZ-GZ 1+1+1 Joint Collaboration Fund No. 4760964. We also gratefully acknowledge the support of SenseTime.

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

APPENDIX

# A IMPLEMENTATION DETAILS

## A.1 NETWORK ARCHITECTURE

We fine-tune all three-stage models starting from pre-trained holistic 3D generators. Specifically, the layout generation model is initialized from TripoSG Li et al. (2025), where we replace the original tokens with our proposed box tokens and inject box ID embeddings to adapt the model for layout generation. Similarly, the coarse structure generation model and the mesh refinement model are initialized using Stage 1 and Stage 2 of TRELLIS, respectively.

In all three stages, we convert half of the DiT blocks to inter-part attention blocks, while the remaining blocks retain intra-part attention. To preserve global semantic context and stabilize fine-tuning, we maintain a holistic generation branch (assigned part ID=0), which helps prevent significant deviation from the pre-trained weights. Furthermore, conditional inputs—such as image or text—are incorporated following the original pre-trained architecture via additional cross-attention blocks. Notably, in each block, all part tokens attend to all condition embeddings.

## A.2 TRAINING AND INFERENCE DETAILS

Owing to GPU memory constraints, we set the maximum number of parts per object to 30 during training. Ror each object sample, we sort its parts by the bounding box sizes and choose the top 30 largest parts during training. Despite this limitation, our framework supports the generation of objects with more than 30 parts through a sequential sampling strategy during inference. The process is as follows: first, we sample an initial set of 30 parts alongside a global part that captures the overall shape. Then, in a subsequent sampling round, we replace the global tokens with the noisy version of the previous denoised global token at each timestep, maintaining the global structure unchanged, and sample new 30 parts. We follow TRELLIS Xiang et al. (2025) to render 24 conditional images for each object and randomly choose one during training. To improve the quality of conditional generation, we adopt common practices from prior work: condition tokens are randomly dropped with a probability of 0.1 during training, and classifier-free guidance with a scale of 3.5 is applied at inference. We also apply a bounding box augmentation strategy in the training of the coarse structure generation model to make the model robust to imperfect bounding box input.

# B PARTVERSE-XL DATASET

To support large-scale, high-fidelity part-aware 3D generation, we present *PartVerse-XL*, a significantly expanded and refined version of the earlier *PartVerse* Dong et al. (2025) dataset. Figure. 8 shows some examples in our dataset. *PartVerse-XL* comprises **40K high-quality 3D objects** sourced from *Objaverse-XL* Deitke et al. (2023), yielding a total of **320K semantically consistent and texture-preserving parts**, each accompanied by a detailed part-level textual description. This scale and diversity—spanning over 200 object categories—substantially surpasses existing part-level benchmarks such as PartNet Mo et al. (2018) and enables robust training of generative models like FullPart that demand both geometric precision and semantic grounding at the part level.

The construction of *PartVerse-XL* follows a two-stage pipeline that combines automated pre-segmentation with rigorous human refinement, ensuring both scalability and annotation quality.

**Automated Pre-segmentation with Semantic Priors.** We begin by leveraging intrinsic modeling cues present in artist-created 3D assets, such as mesh connectivity, UV layout boundaries, and material assignments. To align these low-level cues with high-level semantics, we integrate a 3D-aware segmentation framework built upon SAM-2 Kirillov et al. (2023) and Samesh Tang et al. (2024), enhanced with geometric priors specific to procedural 3D modeling workflows. This hybrid approach produces an initial over-segmented partition of each object, deliberately erring on the side of finer granularity. Over-segmentation is preferred because it provides annotators with atomic building blocks that can be reliably merged, whereas under-segmentation often leads to irreversible loss of part boundaries.

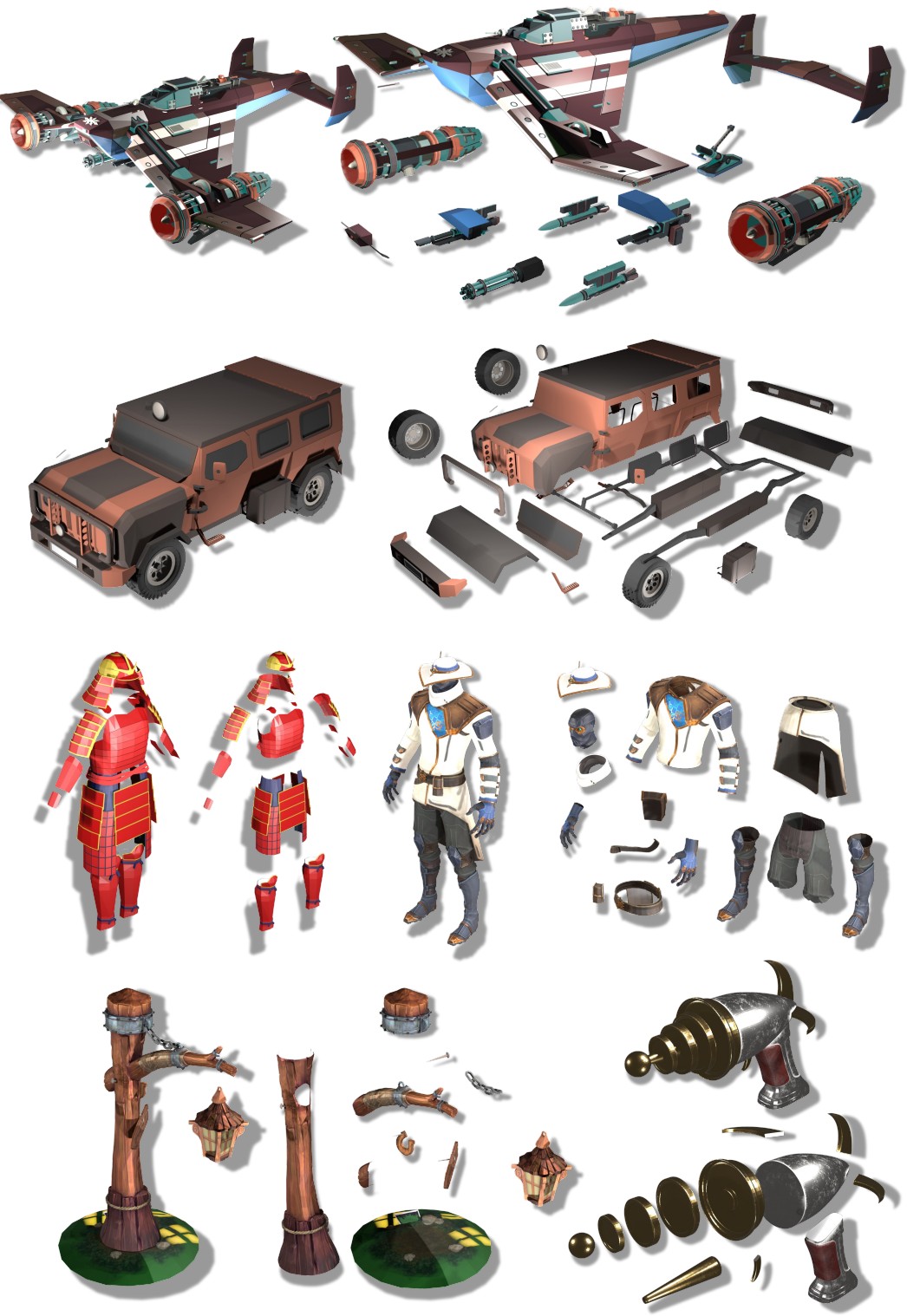

Figure 8: Data examples from PartVerse-XL.

**Human-in-the-Loop Refinement.** All pre-segmented results undergo careful manual curation using a custom Blender-based annotation interface. Annotators first filter out unsuitable assets—such as those with excessive topological complexity, non-manifold geometry, or ambiguous part semantics. They then refine the segmentation by: (1) merging fragments that belong to the same functional or visual unit (e.g., the back and seat of a chair), and (2) splitting regions that conflate distinct components (e.g., separating armrests from chair legs). The refinement protocol emphasizes semantic clarity, structural symmetry, and compatibility with downstream generation tasks. Critically, textures and material properties are preserved throughout this process, ensuring that each extracted part remains visually coherent and renderable.

**Part-Aware Textual Captions.** To enable vision-language conditioning in part-level generation, we generate descriptive captions for every part. For each object-part pair, we render multi-view RGB images of both the full object and the isolated part. We then identify the view that maximizes the visible overlap between the part and its context within the whole object. A composite image is formed by overlaying a bounding box around the part in the full-object render. This image is fed to a state-of-the-art vision-language model (Qwen2.5-VL Wang et al. (2024) in our setting) to produce a natural language description that captures the part's shape, appearance, material, and functional or spatial relationship to the parent object.

*PartVerse-XL* not only provides an order-of-magnitude increase in scale over prior part datasets but also establishes a new standard for semantic consistency, texture fidelity, and multimodal alignment—making it uniquely suited for training and evaluating next-generation part-aware 3D generative models.

## C  MORE RESULTS AND ANALYSIS

### C.1  MEMORY AND EFFICIENCY ANALYSIS

We benchmark the training and inference memory consumption of our model across varying part numbers and present a comparison with prior part generators in Table. 2. All tests are conducted on the same NVIDIA A100 GPU. Common part numbers are typically below 10. PartCrafter and CoPart are incapable of handling objects with more than 16 and 8 parts, due to the fact that their training data was explicitly limited to part counts below these thresholds. The results demonstrate that the memory overhead introduced by our method is acceptable. This efficiency can be attributed to the integration of modern attention mechanisms, specifically FlashAttention, which are highly optimized for processing long sequences. We also adopt gradient checkpointing to reduce memory during training.

Table 2: Memory usage w.r.t. different part counts.

|  | 8 | 16 | 30 |
|---|---|---|---|
| PartCrafter Lin et al. (2025) (infer) | 8.1G | 10.3G | - |
| OmniPart Yang et al. (2025b) (infer) | 12.7G | 13.4G | 14.3G |
| CoPart Dong et al. (2025) (infer) | 42.2G | - | - |
| Ours (infer stage-I) | 7.8G | 10.1G | 14.3G |
| Ours (infer stage-II) | 10.4G | 15.8G | 17.9G |
| Ours (infer stage-III) | 14.1G | 20.3G | 36.3G |
| Ours (train stage-I) | 21.7G | 24.5G | 30.4G |
| Ours (train stage-II) | 23.7G | 28.0G | 35.9G |
| Ours (train stage-III) | 28.2G | 36.2G | 45.0G |

We benchmark the inference latency against existing part generators, as shown in the Table. 3. All tests are conducted on the same NVIDIA A100 GPU. PartCrafter achieves the fastest inference speed while suffering from more artifacts due to its one-stage framework. OmniPart exhibits minimal sensitivity to part quantity due to its shared global coordinate space design, where token count does not increase significantly with the addition of parts. In contrast, our method allocates a full-resolution grid to each part, resulting in a linear growth in token count as the number of parts increases. However, this design choice represents a deliberate trade-off between generation quality and computational efficiency. The dedicated grid space for each part is fundamental to our framework's ability to generate high-fidelity details for small and intricate components—a capability that

shared-space approaches inherently compromise. Despite the linear growth in token count, modern optimization techniques like FlashAttention and efficient memory management prevent latency from scaling proportionally; for instance, with 8 parts (a common configuration for everyday objects), FullPart maintains a practical inference time of **55 seconds**, despite processing several times more tokens than OmniPart. This latency profile remains acceptable for applications prioritizing quality over real-time interaction, such as content creation pipelines and offline asset generation. Future work will explore hierarchical token compression strategies to further improve inference efficiency while preserving our quality advantages.

Table 3: Inference latency w.r.t. different part counts.

|  | 8 | 16 | 30 |
| --- | --- | --- | --- |
| PartCrafter Lin et al. (2025) | 10s | 26s | - |
| OmniPart Yang et al. (2025b) | 17s | 21s | 25s |
| CoPart Dong et al. (2025) | 46s | - | - |
| Ours (stage-I) | 10s | 25s | 42s |
| Ours (stage-II) | 22s | 83s | 181s |
| Ours (stage-III) | 17s | 69s | 131s |
| Ours (stage-I + II + III) | 55s | 187s | 370s |

## C.2 EVALUATION ON PARTNET AND PARTNEXT

To further validate the generalizability of our method, we constructed an additional test set comprising 100 random objects from PartNet Mo et al. (2018) and 100 random objects from PartNeXt Wang et al. (2025). As shown in Table. 4, FullPart consistently outperforms previous methods.

Table 4: Quantitative comparison on PartNet/PartNeXt test set.

|  | F-Score ↑ | CD ↓ | Part-CD ↓ | ULIP-Score ↑ |
| --- | --- | --- | --- | --- |
| TRELLIS Xiang et al. (2025) | 0.75 | 0.15 | - | 0.24 |
| PartCrafter Lin et al. (2025) | 0.55 | 0.51 | - | 0.10 |
| OmniPart Yang et al. (2025b) | 0.76 | 0.13 | 0.46 | 0.20 |
| Ours | 0.77 | 0.12 | 0.42 | 0.25 |

## C.3 MORE QUALITATIVE RESULTS AND COMPARISONS

We show extensive qualitative comparisons with PartCrafter Lin et al. (2025), OmniPart Yang et al. (2025b), and CoPart Dong et al. (2025) in Figure. 9. The results reveal that PartCrafter tends to generate parts that are either coalesced or disjoint, with the latter often appearing as floating artifacts during spatial decoding (e.g., the generated chair and hat have holes and floaters). This issue stems from its use of an implicit vecset representation for direct part generation, where individual part tokens are easily influenced by others, leading to increased artifacts. OmniPart, on the other hand, struggles with fine-grained part generation due to its use of a shared global coordinate space and insufficient spatial resolution. For example, OmniPart omits the surface details of small parts in the first and second results in Figure. 9.

For CoPart, we caption the input image with a VLM Wang et al. (2024) and use the predicted caption to generate parts. The results show that the quality of the textured part mesh generated by CoPart is limited, as it directly employs a holistic 3D generator for refinement without part-specific tuning. In contrast, FullPart achieves superior results through dedicated part-data fine-tuning and a carefully designed part alignment strategy.

We also provide additional in-the-wild qualitative tests in Figure. 10. The qualitative results demonstrate the strong generalization capability of FullPart.

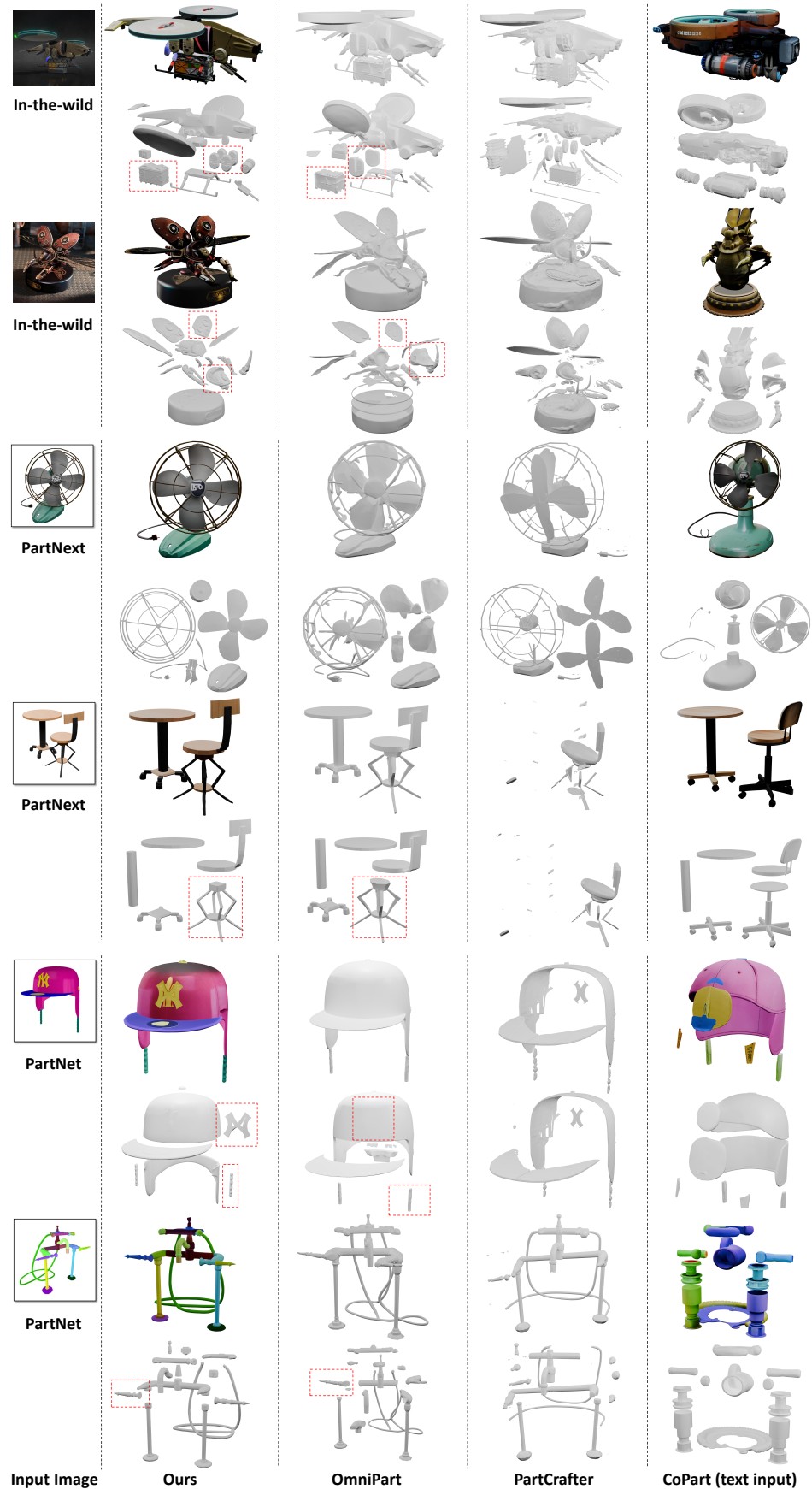

Figure 9: More comparisons with state-of-the-art 3D Part generators.

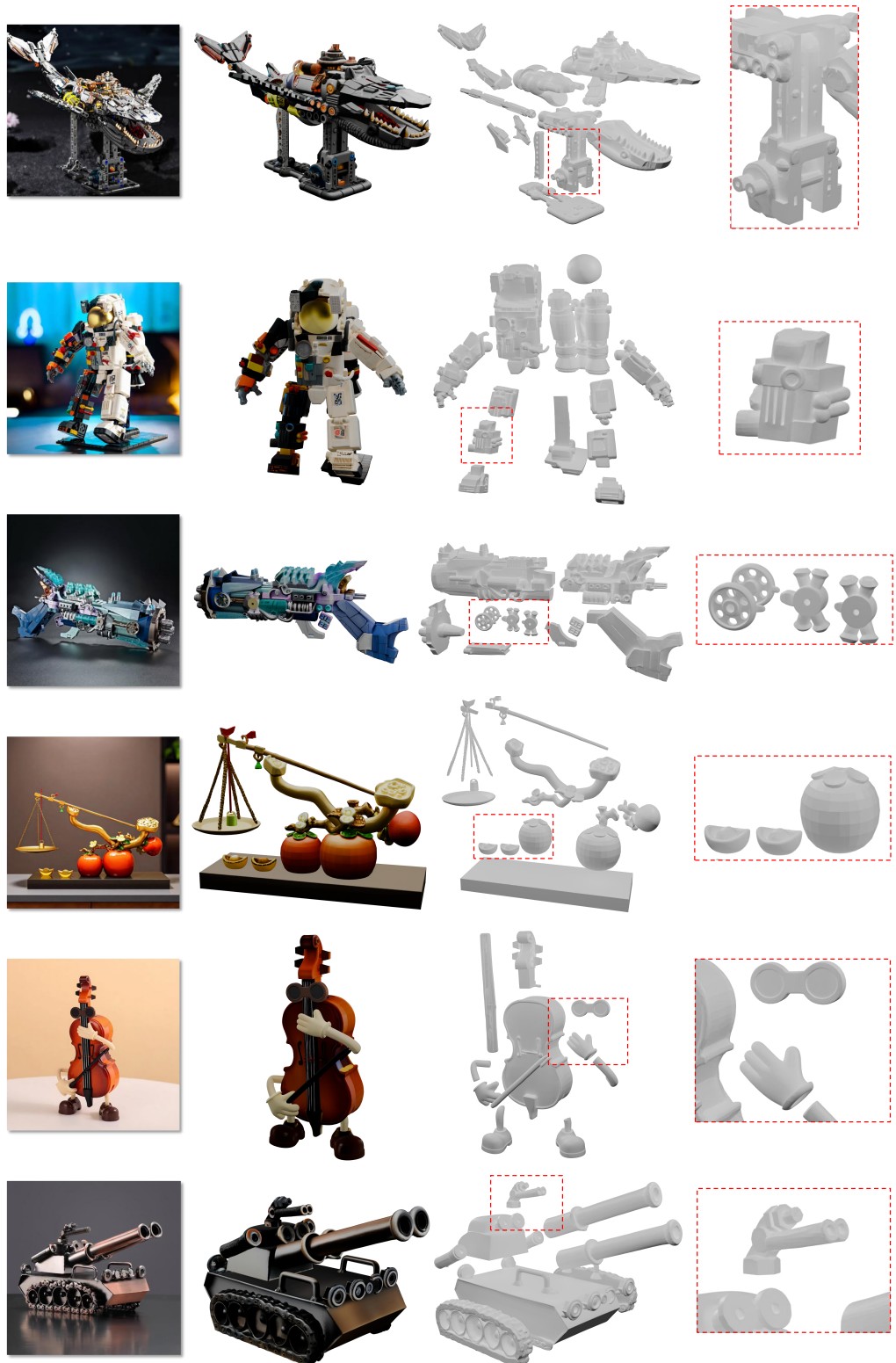

Figure 10: More in-the-wild results of FullPart.

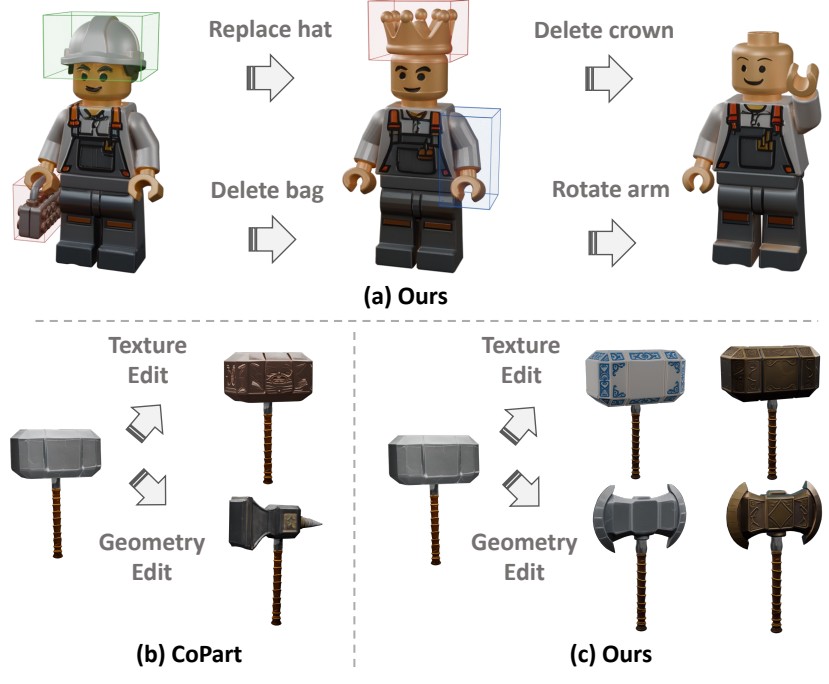

Figure 11: Editing results and comparison.

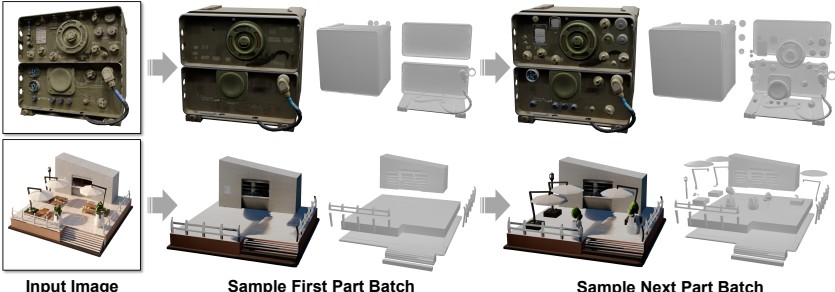

Figure 12: Sequential sampling for super long part sequence generation.

## C.4  LONG SEQUENCE SAMPLING RESULTS

In Figure. 12, we show generation results for objects with over 30 parts using our sequential sampling strategy mentioned in appendix A.2. By maintaining a global branch throughout the process, our approach ensures the coherence and quality of the newly generated parts.

## C.5  MORE DETAILS AND RESULTS OF THE EDITING APPLICATION.

Thanks to our hybrid pipeline, users can edit specific parts by modifying their bounding boxes and re-running the coarse generation and refinement stages. Given an assembled object, we first encode each part into its latent tokens. To regenerate a selected part while preserving others, we leverage the inherent inpainting capability of diffusion models. Specifically, we initialize the target part tokens from Gaussian noise. At each denoising step $t$, we replace the noisy tokens of unchanged parts by adding noise at timestep $t$ to their clean tokens. This creates an inpainting setup where contextual information from the unchanged parts gradually and naturally guides the regeneration of the target part. The regenerated tokens for the selected parts are then inserted or replaced according to the updated box layout. This approach offers significant flexibility. If we want other unselected parts to also change coherently, we can start from a specific step in the sampling process (e.g., t=0.5) and stop replacing the sampled latent with the noised "clean latent". This enables the subsequent denoising process to make subtle adjustments to the other parts. Optionally, we can opt to keep the unselected parts entirely unchanged, achieving perfect preservation of the unmodified regions.

We also demonstrate additional editing operations—including part replacement, rotation, and deletion in Figure. 11 (a). We provide comparisons with prior part-based generative methods Dong et al. (2025) in Figure. 11 (b) and (c). The results confirm that FullPart achieves competitive editing performance.

## C.6 QUANTITATIVE ABLATION RESULTS

We further validate the effectiveness of our designs through quantitative ablation studies. The results in Table. 5 clearly demonstrate the critical importance of the carefully designed center-corner encoding mechanism in enabling the model to accurately perceive the actual size of each part. Moreover, we observe a significant performance drop when directly using part labels from raw meshes without human annotation. Finally, allocating a full-resolution grid to each part is shown to enhance the quality of the generated parts.

Table 5: Quantitative ablation results on PartVerse-XL test set.

| | F-Score ↑ | CD ↓ | Part-CD ↓ | ULIP-Score ↑ |
|---|---|---|---|---|
| w/o Corner Enc | 0.65 | 0.36 | 0.55 | 0.13 |
| w/o Human Anno | 0.72 | 0.20 | 0.48 | 0.17 |
| w/o Per-part Grid | 0.76 | 0.17 | 0.40 | 0.21 |
| Full model | 0.81 | 0.11 | 0.36 | 0.24 |

## C.7 DISCUSSION OF THE LAYOUT GENERATION

Previous methods directly predict the bounding boxes (layout) parameters (x, y, z, w, h, d). For example, LayoutGPT uses LLM to predict box parameters and then retrieve objects based on boxes for 3D scene generation. OmniPart also trains an auto-regressive model to predict part box parameters. In contrast, we regard each bounding box as a box mesh, and fine-tune an implicit vecset diffusion model to generate box meshes and then compute the boxes from the extracted box meshes. In this way, FullPart can inherit the priors from the pre-trained vecset diffusion model and turn the accurate box parameter regression task into a more robust diffusion-based generation process.

## C.8 DIFFERENCE WITH COPART

The first major difference lies in the part representation. CoPart Dong et al. (2025) represents 3D parts using a combination of implicit vecset geometric latents and image latents. In contrast, Full-Part represents each part with explicit voxel-based latents, which allow texture information to be directly encoded into the voxel tokens—similar to TRELLIS. Due to the constraints of its implicit representation, the part geometric tokens in CoPart share a global coordinate space, leading to insufficient resolution for fine-grained parts. In contrast, FullPart assigns each part an independent full-resolution grid, thereby enabling detailed geometric modeling.

Second, the generation stages of CoPart and FullPart differ significantly in their objectives and designs: i) In the first stage, CoPart requires users to provide part bounding boxes as input. These boxes are injected into a dual-branch DiT to generate part-centric multi-view images (via ControlNet) and coarse part shapes (via cross-attention). In contrast, FullPart eliminates the need for manual box input by automatically generating boxes through a box diffusion process. It then employs a compact single-branch DiT to generate coarse voxel-based parts. Notably, thanks to our explicit representation, each voxel latent is intrinsically defined within its corresponding box, eliminating the need for the additional architectural mechanisms used in CoPart to inject box information. ii) In the second stage, CoPart feeds the coarse shapes and multi-view images into a pre-trained holistic 3D object generator to obtain textured part meshes. Notably, CoPart treats each part as an independent whole object and directly employs this holistic generator without any fine-tuning on part-specific data. CoPart claim that the generator can produce textured part meshes when initialized with both coarse shapes and multi-view images. In contrast, FullPart's refinement stage is fine-tuned on part data and incorporates a center-corner encoding mechanism to carefully align different parts, thereby achieving more coherent and higher-quality part generation.

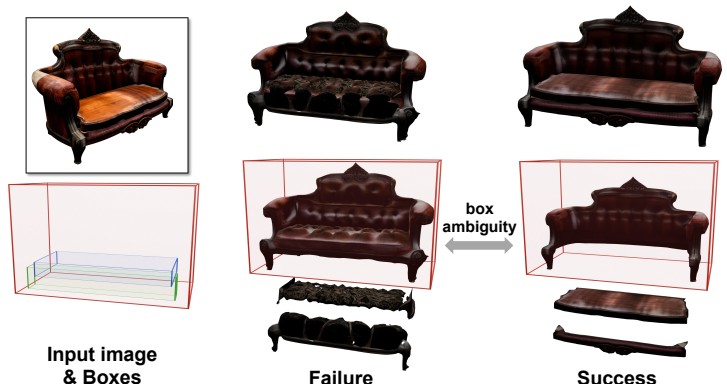

Figure 13: Typical failure case due to box ambiguity.

## C.9 USER STUDY

We conducted a user study involving 73 participants from 10 professional backgrounds. Participants were asked to select the best result based on global coherence or assembly structural plausibility. The results demonstrate that FullPart outperforms all previous methods in both global quality and part-level coherence.

Table 6: User Study (% preference).

| Method | Global coherence | Part structural plausibility |
|---|---|---|
| PartCrafter Lin et al. (2025) | 7.3 | 2.7 |
| OmniPart Yang et al. (2025b) | 38.8 | 21.0 |
| CoPart Dong et al. (2025) | 5.0 | 16.0 |
| Ours | 48.9 | 60.3 |

## D LIMITATIONS AND FAILURE CASE ANALYSIS

We observe two primary limitations. First, the model occasionally generates darker textures, which we attribute to the dark render in the data processing stage: our optimization of render settings for speed came at the cost of quality. Specifically, we lowered critical ray tracing parameters, including the number of light path bounces and the overall sampling rate. These simplifications limited how effectively light could bounce and accumulate in the scene, leading to an underexposed and less accurate image. Second, a typical failure case is shown in Figure. 13. Because the box's structure is ambiguous, the model gets confused about the final design, which results in failures like generating an unnecessary padding layer. In addition, our computational cost increases linearly with the number of parts. We plan to explore ways to accelerate the inference in future work.

## E USE OF LARGE LANGUAGE MODELS

During the writing of this paper, we employed a large language model (GPT-4) solely to improve the clarity and fluency of the original text we had drafted. The model was used as a tool for language polishing and did not contribute to the conceptual development, technical execution, experimental analysis, or scientific conclusions presented in this work

