# OpenReview forum: "FullPart: Generating each 3D Part at Full Resolution"
_ICLR.cc/2026/Conference — ICLR 2026 Poster_

### Official Review · Reviewer_a2W1 · 2025-10-14

**Soundness:** 3
**Presentation:** 3
**Contribution:** 3
**Rating:** 6
**Confidence:** 5

**Summary:**

FullPart: Generating each 3D Part at Full Resolution presents a novel framework for part-based 3D generation that addresses a critical limitation in existing methods: the resolution bottleneck for small parts. The authors identify that prior work, whether using implicit latent tokens or explicit voxel grids, forces all parts to share a single global representation space. This causes small but complex parts to be allocated insufficient resolution, leading to a loss of geometric detail.

Furthermore, the authors introduce PartVerse-XL, a large-scale, high-quality dataset of 40K 3D objects and 320K human-annotated parts, to address the scarcity of reliable training data for part-level generation. Extensive experiments show that FullPart achieves state-of-the-art performance, particularly in generating plausible geometries for small and occluded parts.

**Strengths:**

1. Introduction of a High-Quality, Large-Scale Dataset: PartVerse-XL

The creation and release of PartVerse-XL is a significant contribution that addresses a major roadblock in the field.

Addressing a Critical Data Scarcity: The paper correctly identifies that existing 3D datasets lack high-quality, semantically consistent part-level annotations. Artist-created metadata is often noisy and inconsistent, hindering the training of robust models. The two-stage pipeline of automated pre-segmentation followed by expert human refinement ensures both high quality and semantic consistency. This is a costly but necessary step to create a reliable benchmark. PartVerse-XL provides a foundational resource that will likely accelerate future research in 3D part generation, much like how datasets like ImageNet propelled 2D vision. By training and evaluating on this dataset, FullPart establishes a strong, reproducible baseline for the community.

2. A Well-Designed and Effective Hybrid Pipeline

The core pipeline of FullPart is a thoughtfully engineered solution that effectively combines the best of two worlds. The pipeline is cleverly designed to use the right representation for the right task:

Stage 1 (Implicit for Layout): Bounding boxes contain minimal geometric detail but require an understanding of global structure and inter-part relationships. Implicit diffusion models are perfectly suited for this "conceptual" generation task.

Stages 2 & 3 (Explicit for Geometry): Generating high-fidelity geometry demands precise spatial control. By allocating a dedicated voxel grid to each part, FullPart bypasses the fundamental resolution constraint of shared-grid methods. The ablation study in Figure 6(c) powerfully demonstrates the dramatic improvement this brings, especially for small parts like thin chair legs.

**Weaknesses:**

1. Significant Architectural Similarity to Preceding Work

While the pipeline is effective, its core structure bears a strong resemblance to contemporaneous work, particularly Co-Part (Dong et al., 2025), which may dampen its perceived novelty.

Parallels with Co-Part: Co-Part also proposes a multi-stage, diffusion-based framework for part generation that utilizes a layout stage followed by geometry generation. Both methods employ transformer architectures with intra-part and inter-part attention mechanisms to maintain coherence.


2. Insufficient Qualitative Comparison with Key Part-Based Baselines

The experimental section, while including quantitative metrics, lacks the qualitative depth needed to fully convince the reader of its superiority over other part-aware generators.

Sparse Visual Comparisons: Figure 4 provides only a limited set of visual comparisons against PartCrafter and OmniPart. To truly showcase its advantages, the paper needs a more extensive gallery of side-by-side comparisons, especially on challenging cases.

Failure Cases of Baselines: It would be highly informative to include examples where methods like OmniPart fail dramatically (e.g., a chair where the legs are voxelized into blobs) and show how FullPart succeeds.

Missing Baseline: Given the noted similarity, a direct qualitative comparison with Co-Part is crucial but is absent. Showing a visual comparison would allow for a direct assessment of whether the full-resolution grid leads to tangibly better results than Co-Part's approach.

User Study for Coherence: A user study evaluating the global coherence and structural plausibility of the assembled objects compared to other part-based methods would provide stronger evidence than metrics like Part-CD alone.

3. Under-Exploration of the Editing Application

The editing application presented in Figure 7 is promising but is only briefly mentioned. The paper would be strengthened by a more thorough exploration of this capability. For instance, it could demonstrate more complex edits like part replacement, rotation, or deletion, and quantitatively evaluate the preservation of unedited parts. How does this editing capability compare to other editable part-based generators? A discussion or comparison on this front would highlight a key practical advantage of the proposed pipeline.

**Questions:**

1. Significant Architectural Similarity to Preceding Work
2. Insufficient Qualitative Comparison with Key Part-Based Baselines
3. Under-Exploration of the Editing Application

---

> ### Author Response · Authors · 2025-11-21
> **Response to Reviewer a2W1**
>
> Thank you for your helpful comments. Our replies to your questions and revisions to the submission are stated below.
>
> **W1**: Difference with CoPart.
>
> **A1**:
> When comparing with CoPart, the core distinctions lie in resolution handling, representation design, input automation, and refinement strategy. FullPart fundamentally advances part-based 3D generation through i) per-part full-resolution grids, ii) explicit voxel representations, iii) automated bounding box prediction, and iv) part-specific fine-tuning—collectively enabling higher fidelity and coherence. Below we detail the four key differences:
> ﻿
>
> i) CoPart does not employ full-resolution grids for individual parts; instead, its part geometric tokens share a global coordinate space, inherently limiting detail capacity for fine-grained structures. FullPart addresses this by assigning each part an independent full-resolution voxel grid, enabling high-fidelity geometric modeling without resolution constraints.
> ﻿
>
> ii) CoPart uses an implicit representation combining 3D vector-set geometric latents and separate image latents, necessitating a complex dual-branch DiT architecture (ControlNet for texture, cross-attention for geometry). FullPart adopts an explicit voxel-based representation where geometry and texture are jointly encoded within voxel tokens (similar to TRELLIS). This eliminates CoPart’s architectural complexity, enabling a compact single-branch DiT while improving extensibility—for instance, explicit voxels intrinsically anchor to their bounding boxes without requiring CoPart’s dedicated injection mechanisms.
>
> ﻿
> iii) CoPart requires manual user input of part bounding boxes in its first stage, which are injected into its dual-branch system to guide generation. FullPart eliminates this dependency via a dedicated box diffusion process that auto-generates boxes. This enables end-to-end automation while maintaining geometric precision.
> ﻿
>
> iv) CoPart’s second stage uses a pre-trained holistic 3D generator without fine-tuning on part data, treating each part as an independent object. This yields inconsistent textures and misaligned assemblies. FullPart fine-tunes its entire refinement stage on part-specific datasets and introduces a center-corner encoding mechanism that explicitly models inter-part spatial relationships (e.g., joints and boundaries). This ensures seamless geometric and textural coherence across parts.
>
> **W2**: More comparisons and user study.
>
> **A2**:
> We have added extensive comparisons with PartCrafter, OmniPart, and CoPart in Section C.3 of the supplementary material.
>
> The results reveal that PartCrafter tends to generate parts that are either coalesced or disjointed, with the latter often appearing as floating artifacts during spatial decoding (e.g., the generated chair and hat have holes and floaters). This issue stems from its use of an implicit vecset representation for direct part generation, where individual part tokens are easily influenced by others, leading to increased artifacts. OmniPart, on the other hand, struggles with fine-grained part generation due to its use of a shared global coordinate space and insufficient spatial resolution. For example, OmniPart omits the surface details of small parts in the first and second results in Figure. 9 of the supplementary material.
> For CoPart, we caption the input image with a VLM and use the predicted caption to generate parts. The results show that the quality of the textured part mesh generated by CoPart is limited, as it directly employs a holistic 3D generator for refinement without part-specific tuning. In contrast, FullPart achieves superior results through dedicated part-data fine-tuning and a carefully designed part alignment strategy.
>
> We conducted a user study involving 73 participants from 10 professional backgrounds. Participants were asked to select the best result based on global coherence or assembly structural plausibility. The results demonstrate that FullPart outperforms all previous methods in both global quality and part-level coherence. We have added the user study to Section C.9 of the supplementary material.
>
> **Table 1. User Study (% Preference)**
>
> | **Method**     | **Global Coherence** | **Part Structural Plausibility** |
> |----------------|:--------------------:|:--------------------------------:|
> | PartCrafter    |          7.3           |                 2.7                 |
> | OmniPart       |          38.8           |                 21.0                 |
> | CoPart         |          5.0           |                 16.0                 |
> | **Ours**       |          48.9           |                 60.3                 |

---

> > ### Author Response · Authors · 2025-11-21
> >
> > **W3**: More details and results of the editing application.
> >
> > **A3**:
> > Thanks to our hybrid pipeline, users can edit specific parts by modifying their bounding boxes and re-running the coarse generation and refinement stages. Given an assembled object, we first encode each part into its latent tokens. To regenerate a selected part while preserving others, we leverage the inherent inpainting capability of diffusion models. Specifically, we initialize the target part tokens from Gaussian noise. At each denoising step
> > $t$, we replace the noisy tokens of unchanged parts by adding noise at timestep
> > $t$ to their clean tokens. This creates an inpainting setup where contextual information from the unchanged parts gradually and naturally guides the regeneration of the target part. The regenerated tokens for the selected parts are then inserted or replaced according to the updated box layout.
> > This approach offers significant flexibility. If we want other unselected parts to also change coherently, we can start from a specific step in the sampling process (e.g., t=0.5) and stop replacing the sampled latent with the noised "clean latent." This enables the subsequent denoising process to make subtle adjustments to the other parts. Optionally, we can opt to keep the unselected parts entirely unchanged, achieving perfect preservation of the unmodified regions.
> > We also demonstrate additional editing operations—including part replacement, rotation, deletion, texture editing, and geometry editing; comparisons with prior part-based generative methods are also provided in Section C.5 of the supplementary material.

---

### Official Review · Reviewer_TxpB · 2025-10-27

**Soundness:** 4
**Presentation:** 4
**Contribution:** 3
**Rating:** 8
**Confidence:** 4

**Summary:**

This paper proposes FullPart, a 3D part generation framework integrating implicit and explicit paradigms. It first generates bounding box layouts via implicit box vecset diffusion, then creates each part in an independent full-resolution voxel grid, and uses a center-corner encoding strategy to maintain global coherence. It also introduces PartVerse-XL, a large human-annotated 3D part dataset. Experiments show FullPart achieves state-of-the-art 3D part generation results.

**Strengths:**

1. It identifies the problem that the resolution given to small parts is limited in 3D part generation by explicit voxel representation. And solves this problem by generating each part in an independent full-resolution voxel grid.
2. It creatively proposes a new positional embedding strategy to maintain global coherence in part generation with independent voxel grids.
3. The paper introduces PartVerse-XL, a large human-annotated 3D part dataset, which is a valuable resource for the research community.
4. The figures are clear and attractive, effectively illustrating the framework and results.

**Weaknesses:**

1. It looks like generating each part in full-resolution voxel grids will significantly increase the computational cost of training and inference. However, the paper does not provide any analysis or discussion on the computational efficiency of FullPart compared to existing methods.
2. The evaluation is only conducted on the PartVerse-XL dataset, which is introduced by the authors themselves. It would be more convincing if the authors could also evaluate their method on other established 3D part generation datasets (or synthesize from 3D part understanding datasets) to demonstrate its generalizability.

**Questions:**

1. Is there any innovation in the layout generation stage? Can you introduce how prior work handles layout generation and what are the differences compared to your method?
2. How does the computational cost (in terms of training time, inference time, and memory usage) of FullPart compare to existing 3D part generation methods that use explicit voxel representations?

---

> ### Author Response · Authors · 2025-11-21
> **Response to Reviewer TxpB**
>
> Thank you for your helpful comments. Our replies to your questions and revisions to the submission are stated below.
>
> **W1**: Memory and efficiency analysis.
>
> **A1**:
> Under a common 8-part setting, our peak memory usage (14.1GB) is comparable to the state of the arts (12.7GB), and computational time (55s) is comparable to CoPart (42.2s), and 3x slower than OminPart (17s), but still acceptable.
> We have included both the memory and latency analysis in Section C.1 of the supplementary material.
>
> i) We benchmark the training and inference memory consumption of our model across varying part numbers and present a comparison with prior part generators in the table below.
> All tests are conducted on the same NVIDIA A100 GPU.
> Common part numbers are typically below 10.
> PartCrafter and CoPart are incapable of handling objects with more than 16 and 8 parts, because their training data was explicitly limited to part counts below these thresholds. We also tested extreme cases with a large number of parts (30). The results demonstrate that the memory overhead introduced by our method is acceptable. This efficiency can be attributed to the integration of modern attention mechanisms, specifically FlashAttention, which are highly optimized for processing long sequences. We also adopt gradient checkpointing to reduce memory during training.
>
> **Table 1. Memory Usage (GB) vs. Part Count**
>
> | **Method**          | **8 parts**      | **16 parts** | **30 parts** |
> |---------------------|:---------:|:---------:|:---------:|
> | Ours (train-s₁)     | 21.7          | 24.5      | 30.4      |
> | Ours (train-s₂)     | 23.7         | 28.0      | 35.9      |
> | Ours (train-s₃)     | 28.2          | 36.2      | 45.0      |
> | PartCrafter (infer) | 8.1    | 10.3      |   -   |
> | OmniPart (infer)    | 12.7          | 13.4      | 14.3      |
> | CoPart (infer)      | 42.2          |   -   |   -   |
> | Ours (infer-s₁)     | 7.8    | 10.1      | 14.3      |
> | Ours (infer-s₂)     | 10.4          | 15.8     | 17.9      |
> | Ours (infer-s₃)     | 14.1          | 20.3      | 36.3      |
> | Ours (infer-peak)     | 14.1          | 20.3      | 36.3      |
> >*s₁ / s₂ / s₃ denote different stages.*
>
> ii) We also benchmark the inference latency against existing part generators, as shown in the following table.
> All tests are conducted on the same NVIDIA A100 GPU.
> PartCrafter achieves the fastest inference speed while suffering from more artifacts due to its one-stage framework.
> OmniPart exhibits minimal sensitivity to part quantity due to its shared global coordinate space design, where token count does not increase significantly with the addition of parts. In contrast, our method allocates a full-resolution grid to each part, resulting in a linear growth in token count as the number of parts increases.
> However, this design choice represents a deliberate trade-off between generation quality and computational efficiency. The dedicated grid space for each part is fundamental to our framework's ability to generate high-fidelity details for small and intricate components—a capability that shared-space approaches inherently compromise. Despite the linear growth in token count, modern optimization techniques like FlashAttention and efficient memory management prevent latency from scaling proportionally; for instance, with 8 parts (a common configuration for everyday objects), FullPart maintains a practical inference time of **55 seconds**, despite processing several times more tokens than OmniPart. This latency profile remains acceptable for applications prioritizing quality over real-time interaction, such as content creation pipelines and offline asset generation. Future work will explore hierarchical token compression strategies to further improve inference efficiency while preserving our quality advantages.
>
> **Table 2. Inference Latency (s) vs. Part Counts**
>
> | **Method**                | **8 parts** | **16 parts** | **30 parts** |
> |---------------------------|:-----:|:------:|:------:|
> | PartCrafter               |   10   |   26    |   -    |
> | OmniPart                  |   17   |   21    |   25    |
> | CoPart                    |   46   |   -    |   -    |
> | Ours (s₁)                 |   10   |   25    |   42    |
> | Ours (s₂)                 |   22   |   83    |   181    |
> | Ours (s₃)                 |   17   |   69   |   131    |
> | Ours (s₁ + s₂ + s₃)       |   55   |   187   |   370    |
> >*s₁ / s₂ / s₃ denote different stages.*

---

> > ### Author Response · Authors · 2025-11-21
> >
> > **W2**: More generalizability evaluation.
> >
> > **A2**:
> > To further validate the generalizability of our method, we constructed an additional test set comprising 100 random objects from PartNet and 100 random objects from PartNeXt. As shown in the table below, FullPart consistently outperforms previous methods on this new benchmark. Moreover, we have included additional qualitative results in Section C.3 of the supplementary material. Both quantitative and qualitative results collectively demonstrate the strong generalization capability of FullPart.
> >
> > **Table 3. Quantitative Comparison on PartNet / PartNeXt Test Set**
> >
> > | **Method**     | **F-Score ↑** | **CD ↓** | **Part-CD ↓** | **ULIP-Score ↑** |
> > |----------------|:-------------:|:--------:|:--------------:|:----------------:|
> > | TRELLIS        |       0.75       |    0.15     |       –        |        0.24         |
> > | PartCrafter    |       0.55       |    0.51     |       –        |        0.10         |
> > | OmniPart       |       0.76       |    0.13     |       0.46     |        0.20         |
> > | **Ours**       |       0.77       |    0.12     |       0.42       |        0.25         |
> >
> > **Q1**: Discussion of the layout generation.
> >
> > **A3**: Previous methods directly predict the bounding boxes (layout) parameters (x, y, z, w, h, d). For example, LayoutGPT uses LLM to predict box parameters and then retrieve objects based on boxes for 3D scene generation. OmniPart also trains an auto-regressive model to predict part box parameters. In contrast, we regard each bounding box as a box mesh, and fine-tune an implicit vecset diffusion model to generate box meshes and then compute the boxes from the extracted box meshes. In this way, FullPart can inherit the priors from the pre-trained vecset diffusion model and turn the accurate box parameter regression task into a more robust diffusion-based generation process.
> >
> > **Q2**: Computational cost compared with previous voxel-based part generators.
> >
> > **A4**:
> > Since the training time of OmniPart is unavailable, we compare the inference time and memory usage of FullPart with OmniPart under a common setting of 8 parts. The computational cost of FullPart is acceptable.
> >
> > **Table 4. Computational Cost Compared with OmniPart**
> >
> > | **Method** | **Inference Latency** | **Training Memory (peak)** | **Inference Memory (peak)** |
> > |------------|:----------------------:|:-------------------:|:---------------------:|
> > | OmniPart   |           17s            |         21.4GB           |          12.7GB            |
> > | **Ours**   |           55s            |         28.2GB           |          14.1GB            |

---

### Official Review · Reviewer_YbuT · 2025-10-31

**Soundness:** 3
**Presentation:** 3
**Contribution:** 3
**Rating:** 8
**Confidence:** 3

**Summary:**

This paper presents FullPart, a framework for part-level 3D object generation in which each part is produced at full resolution. The main contribution lies in the integration of implicit and explicit representations:
1. An implicit vecset-based diffusion model learns the layout of parts,
2. An explicit voxel-based representation generates complete high-resolution 3D parts, and
3. The resulting coarse 3D parts are refined into textured meshes.

The authors also introduce PartVerse-XL, a new dataset for part-level 3D generation. Compared with PartCrafter and OmniPart, FullPart demonstrates superior geometric fidelity and structural coherence. Because each part is generated at full resolution, the overall 3D objects contain richer details than those produced by TRELLIS or Direct3D-S2. Ablation studies confirm the effectiveness of Center-Corner Encoding, human-refined annotations, and the Per-Part Full-Resolution Grid.

**Strengths:**

1. The combination of implicit and explicit representations is well-motivated and effectively leverages the strengths of both paradigms.
2. The construction of the PartVerse-XL dataset is a valuable contribution that can advance research in part-level 3D generation.
3. The proposed Center-Corner Encoding mechanism is conceptually simple yet empirically effective.

**Weaknesses:**

1. Section 2.2 discusses mainly recent (2025) part-level 3D generation works. The related work could be more comprehensive by including earlier implicit Tri-plane–based approaches, such as Frankenstein [Yan et al., 2024] and StdGen [He et al., 2024].

**Questions:**

1. The Related Work section could be expanded to discuss a broader range of part-level 3D generation methods, emphasizing the types of representations they employ (e.g., implicit, voxel-based, point-based, or hybrid).
2. Add a section on the use of large language models (LLMs).

---

> ### Author Response · Authors · 2025-11-21
> **Response to Reviewer YbuT**
>
> Thank you for your helpful comments. Our replies to your questions and revisions to the submission are stated below.
>
> **W1**: More part-based generator discussion in related work.
>
> **A1**:
> The Related Work section of the main paper has been revised and now discusses earlier part-based object generators as well as decomposable 3D generators for scenes and characters, covering methods such as Frankenstein [Yan et al., 2024], StdGen [He et al., 2024], and CAST [Yao et al., 2025].
>
> **Q1**: Representation discussion of part level 3D generators in the related work.
>
> **A2**:
> We have revised the related work section to include a broader range of part-level 3D generation methods (object-level, scene-level, and avatars), along with a thorough analysis of the different 3D representations employed in these generative models.
>
> **Q2**: Add a section on the use of large language models (LLMs).
>
> **A3**:
> We have included an explanation of a formal statement regarding the use of the large language model (LLM) for textual refinement in Section E of the supplementary material.

---

### Official Review · Reviewer_2Ufk · 2025-10-31

**Soundness:** 4
**Presentation:** 3
**Contribution:** 3
**Rating:** 8
**Confidence:** 3

**Summary:**

This work introduces FullPart, a novel framework for high-resolution part-aware 3D generation. This framework overcomes the resolution and coherence limitations of previous approaches by combining both implicit and explicit paradigms. Specifically, FullPart consists of three stages: (1) implicit layout generation for spatial arrangement; (2) explicit structure generation at full resolution for fine geometric details; (3) refinement for realistic appearance. Besides, the center-corner encoding mechanism is proposed for global part coherence. To support this framework, this work further introduce PartVerse-XL, the largest human-annotated 3D part dataset with semantically consistent part labels and textual descriptions. Experiments show that FullPart achieves state-of-the-art performance in both part- and object-level evaluations, outperforming existing methods such as PartCrafter, OmniPart, and TRELLIS.

**Strengths:**

1. The proposed per-part full-resolution generation paradigm is technically sound and yields satisfactory results. As shown in Figures 4 and 5, this method can improve the generation performance of both part-level and object-level. Besides, the proposed center-corner encoding mechanism utillizes a unified positional encoding strategy to preserve global spatial coherence across parts of different scales, effectively addressing the spatial discrepancy between different parts.

2. This work presents the largest human-annotated 3D part dataset, including 40K high-quality 3D objects and 320K semantically consistent 3D parts. The scale and diversity of this dataset will facilitate future research on part-aware 3D generation.

3. The paper provides quantitative and qualitative performance comparisons as well as a comprehensive ablation analysis. Experimental results (Table 1, Figure 4, Figure 5) demonstrate the superior performance of FullPart compared to previous part-level and object-level 3D generation methods.

4. The paper is clearly written and well-organized, making it easy to follow.

**Weaknesses:**

1. Generating each part within a full-resolution voxel grid substantially increases memory usage and computational costs, particularly for complex objects with numerous parts. The paper does not include an analysis of computational efficiency or stress testing in this regard.

2. The test set is limited to 100 manually selected untrained objects, which may not comprehensively evaluate the model's generalization capability and overall performance.

3. Quantitative ablation analysis results are missing.

4. This paper lacks a discussion of its limitations and failure cases.

**Questions:**

The sequential sampling strategy for generating over 30 parts appears promising. Could the authors provide results to demonstrate whether the structural coherence or overall geometric quality diminishes as the number of parts increases?

---

> ### Author Response · Authors · 2025-11-21
> **Response to Reviewer 2Ufk**
>
> Thank you for your helpful comments. Our replies to your questions and revisions to the submission are stated below.
>
> **W1**: Memory and efficiency analysis.
>
> **A1**:
> Under a common 8-part setting, our peak memory usage (14.1GB) is comparable to the state of the arts (12.7GB), and computational time (55s) is comparable to CoPart (42.2s), and 3x slower than OminPart (17s), but still acceptable.
>
>  i) We benchmark the training and inference memory consumption of our model across varying part numbers and present a comparison with prior part generators in the table below.
> All tests are conducted on the same NVIDIA A100 GPU.
> Common part numbers are typically below 10.
> PartCrafter and CoPart are incapable of handling objects with more than 16 and 8 parts, because their training data was explicitly limited to part counts below these thresholds. We also tested extreme cases with a large number of parts (30). The results demonstrate that the memory overhead introduced by our method is acceptable. This efficiency can be attributed to the integration of modern attention mechanisms, specifically FlashAttention, which are highly optimized for processing long sequences. We also adopt gradient checkpointing to reduce memory during training.
>
> **Table 1. Memory Usage (GB) vs. Part Count**
>
> | **Method**          | **8 parts**      | **16 parts** | **30 parts** |
> |---------------------|:---------:|:---------:|:---------:|
> | Ours (train-s₁)     | 21.7          | 24.5      | 30.4      |
> | Ours (train-s₂)     | 23.7         | 28.0      | 35.9      |
> | Ours (train-s₃)     | 28.2          | 36.2      | 45.0      |
> | PartCrafter (infer) | 8.1    | 10.3      |   -   |
> | OmniPart (infer)    | 12.7          | 13.4      | 14.3      |
> | CoPart (infer)      | 42.2          |   -   |   -   |
> | Ours (infer-s₁)     | 7.8    | 10.1      | 14.3      |
> | Ours (infer-s₂)     | 10.4          | 15.8     | 17.9      |
> | Ours (infer-s₃)     | 14.1          | 20.3      | 36.3      |
> | Ours (infer-peak)     | 14.1          | 20.3      | 36.3      |
> >*s₁ / s₂ / s₃ denote different stages.*
>
>
> ii) We also benchmark the inference latency against existing part generators, as shown in the following table.
> All tests are conducted on the same NVIDIA A100 GPU.
> PartCrafter achieves the fastest inference speed while suffering from more artifacts due to its one-stage framework.
> OmniPart exhibits minimal sensitivity to part quantity due to its shared global coordinate space design, where token count does not increase significantly with the addition of parts. In contrast, our method allocates a full-resolution grid to each part, resulting in a linear growth in token count as the number of parts increases.
> However, this design choice represents a deliberate trade-off between generation quality and computational efficiency. The dedicated grid space for each part is fundamental to our framework's ability to generate high-fidelity details for small and intricate components—a capability that shared-space approaches inherently compromise. Despite the linear growth in token count, modern optimization techniques like FlashAttention and efficient memory management prevent latency from scaling proportionally; for instance, with 8 parts (a common configuration for everyday objects), FullPart maintains a practical inference time of **55 seconds**, despite processing several times more tokens than OmniPart. This latency profile remains acceptable for applications prioritizing quality over real-time interaction, such as content creation pipelines and offline asset generation. Future work will explore hierarchical token compression strategies to further improve inference efficiency while preserving our quality advantages.
>
> **Table 2. Inference Latency (s) vs. Part Counts**
>
> | **Method**                | **8 parts** | **16 parts** | **30 parts** |
> |---------------------------|:-----:|:------:|:------:|
> | PartCrafter               |   10   |   26    |   -    |
> | OmniPart                  |   17   |   21    |   25    |
> | CoPart                    |   46   |   -    |   -    |
> | Ours (s₁)                 |   10   |   25    |   42    |
> | Ours (s₂)                 |   22   |   83    |   181    |
> | Ours (s₃)                 |   17   |   69   |   131    |
> | Ours (s₁ + s₂ + s₃)       |   55   |   187   |   370    |
> >*s₁ / s₂ / s₃ denote different stages.*
>
> We have included both the memory and latency analysis in Section C.1 of the supplementary material.

---

> > ### Author Response · Authors · 2025-11-21
> >
> > **W2**: More generalizability evaluation.
> >
> > **A2**:
> > To further validate the generalizability of our method, we constructed an additional test set comprising 100 random objects from PartNet and 100 random objects from PartNeXt. As shown in the table below, FullPart consistently outperforms previous methods on this new benchmark. Moreover, we have included additional qualitative results in Section C.3 of the supplementary material. Both quantitative and qualitative results collectively demonstrate the strong generalization capability of FullPart.
> >
> > **Table 3. Quantitative Comparison on PartNet / PartNeXt Test Set**
> >
> > | **Method**     | **F-Score ↑** | **CD ↓** | **Part-CD ↓** | **ULIP-Score ↑** |
> > |----------------|:-------------:|:--------:|:--------------:|:----------------:|
> > | TRELLIS        |       0.75       |    0.15     |       –        |        0.24         |
> > | PartCrafter    |       0.55       |    0.51     |       –        |        0.10         |
> > | OmniPart       |       0.76       |    0.13     |       0.46     |        0.20         |
> > | **Ours**       |       0.77       |    0.12     |       0.42       |        0.25         |
> >
> > **W3**: Quantitative ablation analysis results.
> >
> > **A3**: We have added quantitative ablation results in Section C.6 of the supplementary material and also report here. The results clearly demonstrate the critical importance of the carefully designed center-corner encoding mechanism in enabling the model to perceive the actual size of each part accurately. Moreover, we observe a significant performance drop when directly using part labels from raw meshes without human-annotated data. Finally, allocating a full-resolution grid to each part is shown to enhance the quality of the generated parts.
> >
> > **Table 4. Quantitative Ablation Results on PartVerse-XL Test Set**
> >
> > | **Setting**           | **F-Score ↑** | **CD ↓** | **Part-CD ↓** | **ULIP-Score ↑** |
> > |-----------------------|:-------------:|:--------:|:--------------:|:----------------:|
> > | w/o Corner Enc        |       0.65       |    0.36     |       0.55        |        0.13         |
> > | w/o Human Anno        |       0.72       |    0.20     |       0.48        |        0.17         |
> > | w/o Per-part Grid     |       0.76       |    0.17     |       0.40        |        0.21         |
> > | **Full model**        |       0.81       |    0.11     |       0.36        |        0.24         |
> >
> > **W4**: Discussion of limitations and failure cases.
> >
> > **A4**:
> > We have added the discussion of limitations and failure case analysis to Section D of the supplementary material. We observe two primary limitations.
> > First, the model occasionally generates darker textures, which we attribute to the dark render in the data processing stage: our optimization of render settings for speed came at the cost of quality. Specifically, we lowered critical ray tracing parameters, including the number of light path bounces and the overall sampling rate. These simplifications limited how effectively light could bounce and accumulate in the scene, leading to an underexposed and less accurate image.
> > Second, a typical failure case is shown in Section D of the supplementary material. Because the box's structure is ambiguous, the model gets confused about the final design, which results in failures like generating an unnecessary padding layer.
> > In addition, our computational cost increases linearly with the number of parts. We plan to explore ways to accelerate the inference in future work.
> >
> > **Q1**: Long sequence sampling results.
> >
> > **A5**:
> > We provide generation results for objects with over 30 parts using our sequential sampling strategy in Section C.4 of the supplementary material. By maintaining a global branch throughout the process, our approach ensures the coherence and quality of the newly generated parts.

---

### Author Response · Authors · 2025-11-21

We express our sincere appreciation to the reviewers for their efforts and valuable feedback. All of your suggestions have been incorporated into the revised versions of the main paper and supplementary material, which are all highlighted in magenta.

---

### Author Response · Authors · 2025-12-02

Dear Area Chair,

We thank you and the reviewers for their thoughtful evaluations. Below we concisely summarize our main contributions, the reviewers’ positive feedback, and how our rebuttal and revisions address the key concerns.

---

### Main Contributions and Positive Feedback

Our main contributions are summarized below:

* **New 3D representation and hybrid pipeline**.
  FullPart introduces a hybrid implicit–explicit framework: an implicit vecset diffusion model for part layouts (bounding boxes), followed by explicit voxel-based stages for detailed geometry and texture. This design leverages global structural priors while enabling precise geometric control. Reviewers `YbuT` and `a2W1` find this combination well-motivated and effective.

* **Network design for coherent, part-aware generation**.
  We generate **each part in its own full-resolution voxel grid**, rather than letting all parts share a single global coordinate. In shared-space designs, small parts are squeezed into only a few voxels and easily become blurry or merged; per-part grids remove this bottleneck and noticeably improve small-part details. An effective **Center–Corner Encoding** further encodes part centers and corners jointly to preserve global spatial coherence across parts of different scales. Reviewers `2Ufk` and `TxpB` consider this paradigm technically sound and highlight the effectiveness of Center–Corner Encoding.

* **Generation quality**.
  Experiments show that FullPart improves both part-level and object-level quality over prior/simultaneous methods, and produces visually coherent assemblies with rich local details. In the rebuttal, we further support this with a user study of 73 participants, where FullPart is most preferred in terms of global coherence and structural plausibility. FullPart also supports flexible part-level editing (replacement, rotation, deletion, geometry and texture edits) via diffusion inpainting while preserving unedited parts.

* **New large-scale dataset: PartVerse-XL**.
  We construct **PartVerse-XL**, a human-annotated dataset of **40K objects / 320K parts** with semantically consistent part labels, serving as both training data and a benchmark for part-aware 3D generation and editing. **All four reviewers** recognize this as a valuable, high-quality resource that addresses the scarcity of part-level 3D data and can benefit the community.

Additionally, reviewers note that the paper is **clearly written and well-organized**, with clear figures and strong quantitative and qualitative results (`2Ufk`, `TxpB`, `a2W1`). Three reviewers give a rating of **8** and one gives **6**.

---

### Reviewers’ Concerns and Our Responses

Regarding the reviewers’ concerns, we conducted additional experiments and analyses:

* **Computational efficiency** (`2Ufk`, `TxpB`).
  We include new experiments showing that FullPart’s memory cost is comparable to baselines and its runtime is acceptable for offline pipelines. Under a common 8-part setting, FullPart’s peak inference memory (14.1 GB) is close to the concurrent OmniPart (12.7 GB), and its inference is acceptable and reasonable given the substantial gains in fine-grained detail and user-perceived quality.

* **Generalization beyond PartVerse-XL** (`2Ufk`, `TxpB`).
  We demonstrate that FullPart generalizes well to **external datasets** and **in-the-wild inputs**. On an additional benchmark of 100 PartNet + 100 PartNeXt objects, FullPart consistently outperforms TRELLIS, PartCrafter, and OmniPart on standard geometric and semantic metrics. We also showcase more qualitative comparisons and results on in-the-wild inputs (appendix Fig.9, Fig.10).

* **Difference from CoPart and benefits** (`a2W1`).
  We clarify that FullPart not only differs architecturally from CoPart but also brings concrete advantages: (1) **Per-part full-resolution grids** (vs. CoPart’s shared global space) enable much sharper small-part details; (2) **Explicit voxel tokens with joint geometry–texture encoding** (vs. CoPart’s dual-branch implicit design) simplify the architecture and make spatial reasoning more direct; (3) **Automatic box diffusion** eliminates the need for user-provided bounding boxes, enabling a fully automatic pipeline; and (4) **Part-specific refinement with Center–Corner Encoding** improves cross-part coherence over CoPart’s holistic generator.

* **Additional analyses and clarifications** (`2Ufk`, `YbuT`, `TxpB`, `a2W1`).
  We further strengthen the paper by adding quantitative ablations, more detailed and diverse editing results (appendix C.5), examples with over 30 parts (appendix C.4), and a limitations/failure-case discussion. We also expand the related-work/representation discussion and formally clarify that LLMs are used only for textual refinement. All new numerical results, visualizations, and explanations are included in the revised manuscript and appendix.

---

We respectfully hope this summary helps your assessment of the paper.

Sincerely,

The authors

---

### Meta-Review · Area_Chair_nEK5 · 2026-01-06

**Summary:**

This paper introduces FullPart, a part-level 3D generation framework that combines implicit and explicit generation. The method first generates a part layout by predicting a set of part bounding boxes using an implicit diffusion process over box vectors. Given the predicted layout, FullPart then generates high-fidelity geometry for each part in its own fixed, full-resolution voxel grid. Unlike prior approaches that generate all parts within a shared low-resolution global space, FullPart keeps every part, even small ones, at full resolution, enabling finer geometric details. To maintain consistency across parts of different sizes, the paper proposes a center-point encoding strategy that helps align information exchanged between parts, mitigating misalignment and improving global coherence. In addition to the framework, the paper presents PartVerse-XL, a large-scale human-annotated 3D part dataset containing 40K objects and 320K parts, designed to address the lack of reliable part-level supervision and support training at scale.

**Reviewer Concerns:**

Reviewers mostly raised concerns about the lack of analysis of computational cost and memory usage, as well as the evaluation setup. In particular, the evaluation concerns seem crucial.

- The authors used only 100 shapes for testing while using 40K shapes for training. They could instead split the full dataset (e.g., 80/20) for train/test. Although they also evaluated on 200 additional shapes from other datasets, this is still a very small scale compared to the size of their dataset, most of which is used for training.

Additionally,

- Despite the fact that this is a 3D generation task, the authors do not follow standard evaluation protocols for 3D generative models, instead reporting only 3D reconstruction loss. For instance, TripoSG reports Normal-FID and GPTEval3D, and TRELLIS reports FID based on PointNet++ features, and also reporting CLIP, FID, and KID computed on rendered images. These two methods are used as baselines in this paper, yet the authors do not adopt the evaluation protocols from those works, nor do they explain why.

- Moreover, the authors do not clearly state whether the baseline methods were trained on the same training dataset. This is essential for a fair comparison, since the authors use the 40K training shapes they created. The proposed method could benefit substantially from the scale of the training data.

Reviewers also raised concerns about (1) a lack of quantitative analysis in the ablation study, (2) missing comparisons with an important baseline method (CoPart), and (3) insufficient qualitative results.

For (1), the authors provided additional results. For (2) and (3), they provided only five additional qualitative comparisons and no quantitative comparison with CoPart.

**Reviewer Scores:**

This paper received scores of 8, 8, 8, and 6. However, the AC believes the submission needs to go through a major revision due to the following three issues. (1) the lack of 3D reconstruction evaluations consistent with prior work, (2) insufficient qualitative comparisons, and (3) the small test set size.

The SAC thinks the paper introduces a new task that can inspire future research. These three issues are not significant (as it is a new task) and can be addressed in the revision. The decision is to accept the paper.

---

### Decision · Program_Chairs · 2026-01-26

Accept (Poster)